# 3D imaging of Sox2 enhancer clusters in embryonic stem cells

Zhe Liu[1,2]*, Wesley R Legant[3], Bi-Chang Chen[3], Li Li[3], Jonathan B Grimm[3], Luke D Lavis[3], Eric Betzig[3], Robert Tjian[2,4]

[1]Junior Fellow Program, Howard Hughes Medical Institute, Janelia Research Campus, Ashburn, United States; [2]Transcription Imaging Consortium, Howard Hughes Medical Institute, Janelia Research Campus, Ashburn, United States; [3]Howard Hughes Medical Institute, Janelia Research Campus, Ashburn, United States; [4]LKS Bio-medical and Health Sciences Center, University of California, Berkeley, Berkeley, United States

**Abstract** Combinatorial cis-regulatory networks encoded in animal genomes represent the foundational gene expression mechanism for directing cell-fate commitment and maintenance of cell identity by transcription factors (TFs). However, the 3D spatial organization of cis-elements and how such sub-nuclear structures influence TF activity remain poorly understood. Here, we combine lattice light-sheet imaging, single-molecule tracking, numerical simulations, and ChIP-exo mapping to localize and functionally probe Sox2 enhancer-organization in living embryonic stem cells. Sox2 enhancers form 3D-clusters that are segregated from heterochromatin but overlap with a subset of Pol II enriched regions. Sox2 searches for specific binding targets via a 3D-diffusion dominant mode when shuttling long-distances between clusters while chromatin-bound states predominate within individual clusters. Thus, enhancer clustering may reduce global search efficiency but enables rapid local fine-tuning of TF search parameters. Our results suggest an integrated model linking cis-element 3D spatial distribution to local-versus-global target search modalities essential for regulating eukaryotic gene transcription.

*For correspondence: liuz11@janelia.hhmi.org

## Introduction

The existence and importance of long-range interactions between distal cis-control elements and cognate core promoter factors in regulating gene expression programs that govern cell-fate in animals have been extensively studied in traditional biochemistry, genetics, and genomics (*Levine and Tjian, 2003*; *Levine et al., 2014*). However, these earlier classical studies were unable to capture the three dimensional (3D) spatial organization or the temporal dynamics of the functional interactions between sequence-specific transcription factors (TFs) and distal enhancers. The more recent development of Chromosome Conformation Capture (3C) and high throughput sequencing based techniques have provided additional insights into long-distance chromatin looping, genome folding, and topological domains in the context of whole animal genomes but without providing direct spatial information (*Dostie et al., 2006*; *Lieberman-Aiden et al., 2009*; *Dixon et al., 2012*; *van de Werken et al., 2012*). Indeed, emerging evidence suggests that proximity ligation frequency based distances measured by 3C assays may be limited in its capacity to accurately capture 3D molecular proximity (*Gavrilov et al., 2013*; *O'Sullivan et al., 2013*; *Belmont, 2014*). The inherent constraints of using fixed cells or population based assays to dissect the nature of 3D enhancer organization and transcription factor search dynamics can, however, be partly overcome by single live-cell imaging. Recent advances in fluorescence super resolution microscopy and protein labeling chemistry make possible the visualization and tracking of individual transcription factors as they diffuse and bind to specific targets in the nucleus of living mammalian cells (*Mazza et al., 2012*; *Gavrilov et al., 2013*; *Izeddin et al., 2014*; *Chen et al., 2014b*).

**eLife digest** Stem cells in an embryo have the potential to become any type of cell in the body. When a cell begins to specialize, it loses this ability and can only become a limited number of cell types. These transitions are caused by changes in gene expression. Proteins called transcription factors bind to DNA to switch different genes on or off as cells become specialized.

One such transcription factor, called Sox2, binds to particular DNA sequences in the cell's nucleus to encourage nearby genes to be expressed at the right levels and keep a stem cell unspecialized. However, how these binding sites are positioned throughout the three-dimensional space inside the nucleus was unknown, as was the likelihood of Sox2 finding and binding to these sites.

Now Liu et al. have taken advantage of advanced microscopes to observe the interaction between Sox2 and its binding site in the nucleus of living embryonic cells. This three-dimensional imaging technology is powerful enough to capture images of individual molecules; and Liu et al. attached fluorescent tags to Sox2 to make it easier to watch them in action. By making a series of time-lapse movies, it was revealed that instead of being evenly scattered in the nucleus, Sox2's binding sites are grouped together to form individual clusters; these clusters preferably occupy spaces in the nucleus that are likely enriched for active genes.

Liu et al. suggest that the clustering of Sox2 binding sites makes it more difficult for a Sox2 protein to find these sites at first, but much easier to find when the Sox2 protein is near to the cluster. Thus, the uneven positioning of the binding sites for transcription factors may provide an additional layer of control over gene expression. In the future, it would be important to map Sox2's binding sites while visualizing the activities of single genes in living cells. This would improve our understanding of how the structural organization of the contents of the nucleus can influence the correct timing of specific patterns of gene expression.

If specific and stable TF:DNA binding events can be localized and visually reconstructed at single-molecule resolution within an intact nucleus, we would have an opportunity to map and decipher critical spatial features linked to the 3D organization of the functional genome and simultaneously measure differences in the dynamic nature of the TF target search process in distinct compartments within living cells.

In our recent work (*Chen et al., 2014b*), we described a single-cell, single-molecule imaging strategy to study the in vivo Sox2 and Oct4 target search process and dissect the kinetics of enhanceosome formation at endogenous single-copy gene loci in live embryonic stem (ES) cells. We found that Sox2 and Oct4 search for their cognate targets via a trial-and-error mechanism in which these two TFs undergo multiple rounds of diffusion and non-specific chromatin collisions before stably engaging with a specific target via an ordered assembly mechanism. Single-molecule in vitro measurements indicate that Sox2 can also slide along short stretches of naked DNA to search for its target. Although our findings revealed significant mechanistic insights of the in vivo TF target search process, these initial single molecule tracking (SMT) studies were constrained to investigate the average behavior of TF dynamics in single cells. We were not able to address whether TFs behave differently within distinct sub-nuclear territories such as active gene enriched euchromatic regions vs the more tightly compacted regions of heterochromatin nor whether the 3D spatial distribution of enhancer sites might affect target search dynamics.

To develop new approaches to probe 3D genome organization and address some of these important unresolved questions regarding the dynamic TF target search process, here we took advantage of further developments in super resolution microscopy (*Chen et al., 2014a*) and fluorescent dye chemistry (*Grimm et al., 2015*). We applied lattice light-sheet single-molecule imaging to selectively localize, track, and map endogenous Sox2 binding sites in single, living ES cells. Two-color imaging enabled us to quantify the spatial distribution of Sox2 binding sites (enhancers) with respect to euchromatic vs heterochromatic regions. We also measured potential differential rates of Sox2 diffusion and binding modes within enhancer clusters compared to heterochromatic regions. SMT and Monte Carlo simulations of the Sox2 target search process revealed two distinct behaviors—a 3D diffusion dominant long-range mode when traveling between clusters and a local binding dominant search mode within individual binding clusters. These studies suggest that enhancer clustering may reduce global target

search efficiency but enable rapid local fine-tuning of search parameters that govern spatially controlled gene regulation in the nucleus. We also probed potential links between enhancer clustering and epigenetic regulation. Together, these results reveal principles that integrate 3D enhancer organization with dynamic in vivo TF-DNA interactions that may play a key role in regulating stem cell pluripotency. The combination of methods described here also open new avenues for studying single live-cell genome spatial organization and function.

## Results

### 3D localization of stable Sox2 binding sites in live ES cells

Although numerous studies have been conducted to investigate Sox2:enhancer interactions by biochemical and genomic approaches, no direct sub-nuclear global spatial information of Sox2 enhancer sites has been attained. This aspect of dissecting TF function presents a particular challenge, because the majority of Sox2 molecules (>74%) in the nucleus are in a dynamically diffusing state (*Kaur et al., 2013*; *Chen et al., 2014b*). Our recent single-molecule tracking (SMT) experiments found that Sox2 interactions with DNA consist of two distinct populations: non-specific collisions of short duration (residence time ~0.7 s) and specific 'stable' interactions of much longer duration (residence time ~12 s) (*Chen et al., 2014b*). Since only ~3% of the Sox2 molecules in the nucleus are bound to specific DNA sites at a given window of time, it is impossible to infer the spatial distribution of Sox2 enhancer sites simply from fluorescence fluctuations captured by wide-field imaging or from conventional super resolution images of live or fixed cells.

Currently, the only information we have that can distinguish site-specific binding from non-specific binding events or rapidly diffusing molecules is the relatively long specific residence times of Sox2 at putative cognate recognition sites (*Chen et al., 2014b*). Therefore, we set out to devise a time-resolved, live-cell imaging strategy to selectively localize, track, and map these longer lived 'stable' Sox2 binding events that likely represent site specific Sox2 binding events to generate a super resolution 3D Sox2/enhancer site map for the whole nucleus. To achieve this, we implemented a lattice light-sheet based single-molecule imaging strategy (*Figure 1A*, see *Figure 1—figure supplement 1A–B* for details of optical layout). We first used an improved labeling method in which a HaloTag ligand based on a newly developed fluorophore, Janelia Fluor 549 (JF549) (*Grimm et al., 2015*), at ultralow concentrations (~0.1 fM) was gradually diffused into HaloTag-Sox2 expressing ES cells to fluorescently tag individual Sox2 molecules. During the labeling, we performed iterative cycles of z tiling by light-sheet microscopy of ES cell nuclei that allowed us to image Sox2 at single molecule resolution in 3D in a series of time-lapse movies. Background fluorescence contributed by rapidly diffusing free JF549-HaloTag ligand was negligible under these imaging conditions as single molecules were only detectable inside cell nuclei but not in the cytoplasm or other regions lacking Sox2 binding sites (*Video 1*). Light-sheet imaging turned out to be critical for the success of this strategy because the selective plane illumination not only preserved the photon budget by preventing out-of-focus molecules from photo-bleaching but also significantly increased the signal-to-noise ratio. With 3D localization at high precision (xy: 14 nm, z: 34 nm, *Figure 1—figure supplement 1C*, *Video 2*) coupled to single molecule tracking, we were able to selectively preserve the global positions where single Sox2 molecules dwell (<50 nm) for at least 3 s. The average residence time of selected molecules was ~6.92 ± 0.51 s (n = 9 cells) (*Figure 1B*), consistent with the notion that most of these events likely reflect the longer residence times representing specific Sox2-enhancer interactions (*Chen et al., 2014b*). We next calculated the number of local neighbors for each Sox2 enhancer site to generate a color-coded heat map for visualizing this data (*Figure 1C* and *Video 3*). As can be seen in *Figure 1C*, many local density hot spots were observed within a single nucleus, suggesting that instead of being uniformly distributed throughout the nucleus, Sox2 bound enhancers form locally enriched distinct higher density clusters (EnCs).

### A star burst distribution of Sox2 enhancers in the nucleus

To test whether the clustering behavior of stable Sox2 binding sites was due to potential artifacts introduced by our imaging strategy, we also inspected ES cells that stably expressed a control HaloTag fusion protein, the histone subunit (HaloTag-H2B) using the same imaging set-up followed by an identical computational pipeline and presentation scheme (*Figure 2A* and *Video 4*). In contrast to Sox2, we observed dramatically decreased clustering behavior of HaloTag-H2B (*Figure 2A* and *Video 4*). In order to establish a more quantitative description of the Sox2-enhancer clustering behavior, we

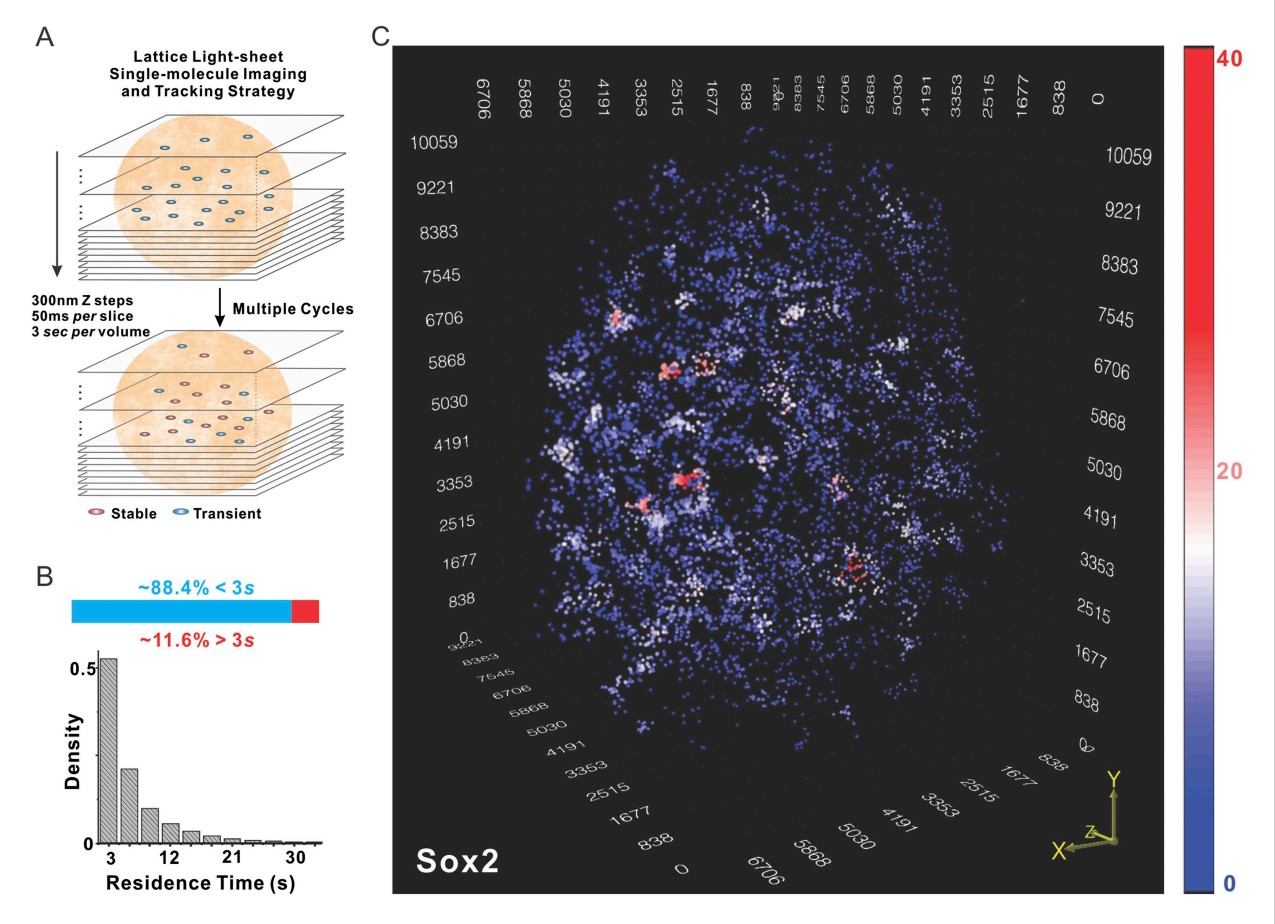

**Figure 1**. Localization of Sox2 stable binding sites in 3D by lattice light-sheet, single-molecule imaging. (**A**) Whole-nucleus single molecule imaging was performed by lattice light-sheet microscopy with 300 nm z steps and 50 ms per frame. HaloTag-Sox2 molecules were labeled by membrane permeable JF549 dye. The imaging scheme was cycled every 3 s for ~500 times. The 3D positions of single molecule localization events were tracked (for more details, see 'Materials and methods'). Any Sox2 molecules that dwelled at a position for more than 3 s were counted as stable bound events. See **Videos 1 and 2** for the exemplary raw data. (**B**) Upper: out of total localized and tracked events, only ~11.6 ± 3.2% had residence times longer than 3 s. ~88.4 ± 6.5% Sox2 molecules appeared in single frames (n = 9 cells). Lower: residence time histogram of stable bound Sox2 molecules. The average residence time detected by this imaging set-up is ~6.92 ± 0.51 s (n = 9 cells). (**C**) 3D density map of stable Sox2 binding sites in single ES cell nucleus. For fair comparisons between experimental conditions, we only considered 7000 stable binding sites for each cell. The color map reflects the number of local neighbors that was calculated by using a canopy radius of 400 nm. The unit of the x, y, z axes is nm. See **Video 3** for the full 3D rotation movie.

The following figure supplement is available for figure 1:

**Figure supplement 1**. Optics layout, PSF, and localization uncertainty estimation.

adapted a pair-correlation function used by cosmologists to describe the clustering behavior of stars in galaxies (**Peebles, 1973**; **Peebles and Hauser, 1974**) (**Figure 2B**, see details in **Equations 3–7**). Briefly, the pair correlation function, *g(r)*, describes the density of spots in a volume element at a separation r from single spots relative to the average density in the whole volume. If enhancer sites were uniformly distributed (**Figure 2—figure supplement 1A** and **Video 5**) the pair correlation function would equal 1 (**Figure 2B** and **Figure 2—figure supplement 1D**, gray diamonds) because the local densities around each position would be invariant and equal to the average density in the entire volume. However, when spots are highly clustered, the *g(r)* will start with values much greater than 1 and gradually decrease as *r* increases, indicating that the local molecular densities around individual spots would be much higher than the average density in the volume. As expected, the *g(r)* function of Sox2 stable binding sites agreed well with a highly clustered behavior while by contrast, the *g(r)* function of H2B suggests a much more random and uniform distribution in the nucleus (**Figure 2B**). We next

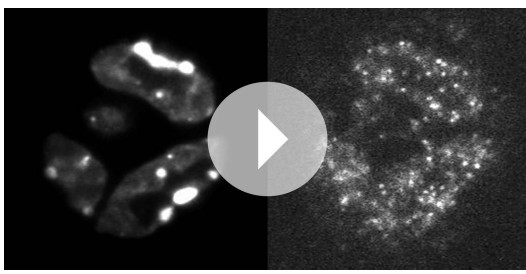

**Video 1**. Single-molecule light-sheet imaging of Sox2 in GFP-HP1 ES cells. HaloTag-Sox2 is gradually labeled with JF549 ligand by diffusion. Light-sheet imaging was performed with a z step of 200 nm.

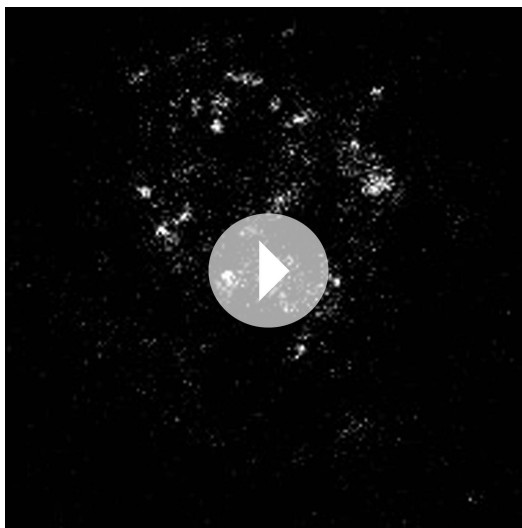

**Video 2**. Single-molecule, light-sheet imaging of HaloTag-Sox2 in single live ES cells. The z step size is 300 nm.

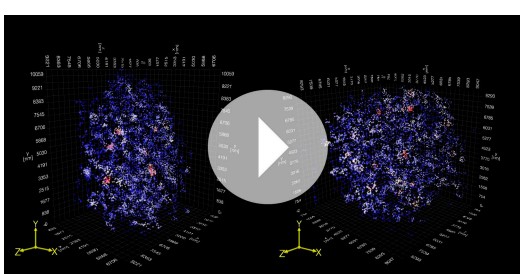

**Video 3**. Reconstructed Sox2 stable binding sites in the live ES cell nucleus. HaloTag-Sox2 stable binding sites (7000, >3 s) were localized, tracked, and reconstructed with a color map same as *Figure 1C*. The unit is nm. 2 cells were shown here.

extended the previously established fluctuation model for describing two dimensional heterogeneous protein distribution in membranes (*Sengupta et al., 2011*) to fit the $g(r)$ function calculated from our 3D dataset (*Equations 10–13*). This model extracted two key parameters related to molecular clustering: the fluctuation range ($\varepsilon$) and the fluctuation amplitude ($A$) (*Supplementary file 1*). Specifically, $\varepsilon$ is proportional to the average size of clusters while $A$ is proportional to the relative molecular density within clusters. We observed, on average, a 14 fold higher fluctuation amplitude of Sox2-enhancers compared with those of H2B. However, we did observe a certain degree of H2B density fluctuations at much larger scales (*Supplementary file 1*), probably reflecting chromatin density variations in the nucleus as reported previously (*Young et al., 1986*). Because we use the 7000 most stable H2B spots to calculate the pair-correlation functions, according to Nyquist sampling theorem, our results are more sensitive to large-scale H2B density fluctuations in the nucleus and may overlook smaller-scale local H2B clustering. The mathematic tools established here should also serve as the basis for future comparisons when we carry out perturbation experiments that will be instructive for dissecting the function and molecular mechanisms underlying enhancer clustering. To determine whether the blinking of stably bound fluorescently tagged Sox2 molecules might influence or distort the observed 'stable' binding of Sox2 in the clusters, we plotted the number of detected events as a function of frame number. These plots show an initial decay that eventually reaches a plateau (*Figure 2—figure supplement 2D*). Such a temporal decay profile is more consistent with a bleaching dominant mechanism in which an equilibrium has been achieved between photobleaching and the ongoing fluorescent labeling of HaloTag-Sox2 molecules. Perhaps the strongest argument that the Sox2 clustering pattern we observe is not likely an artifact of the imaging modality can be derived from the fact that chromatin bound HaloTag-H2B molecules using precisely the same imaging strategy failed to show such a prominent clustering pattern.

To test the contribution, if any, of non-specific interactions to the dramatic clustering behavior observed for Sox2 long-lived binding sites within the cell, we also investigated the clustering behavior of shorter-lived (<3 s) Sox2 binding sites that were initially filtered out in our mapping experiments (*Figure 1B*). If the recorded Sox2 stable binding events mainly reflect random non-specific interactions, the clustering behavior of shorter lived binding sites should be similar to that observed for the long lived putative 'specific'

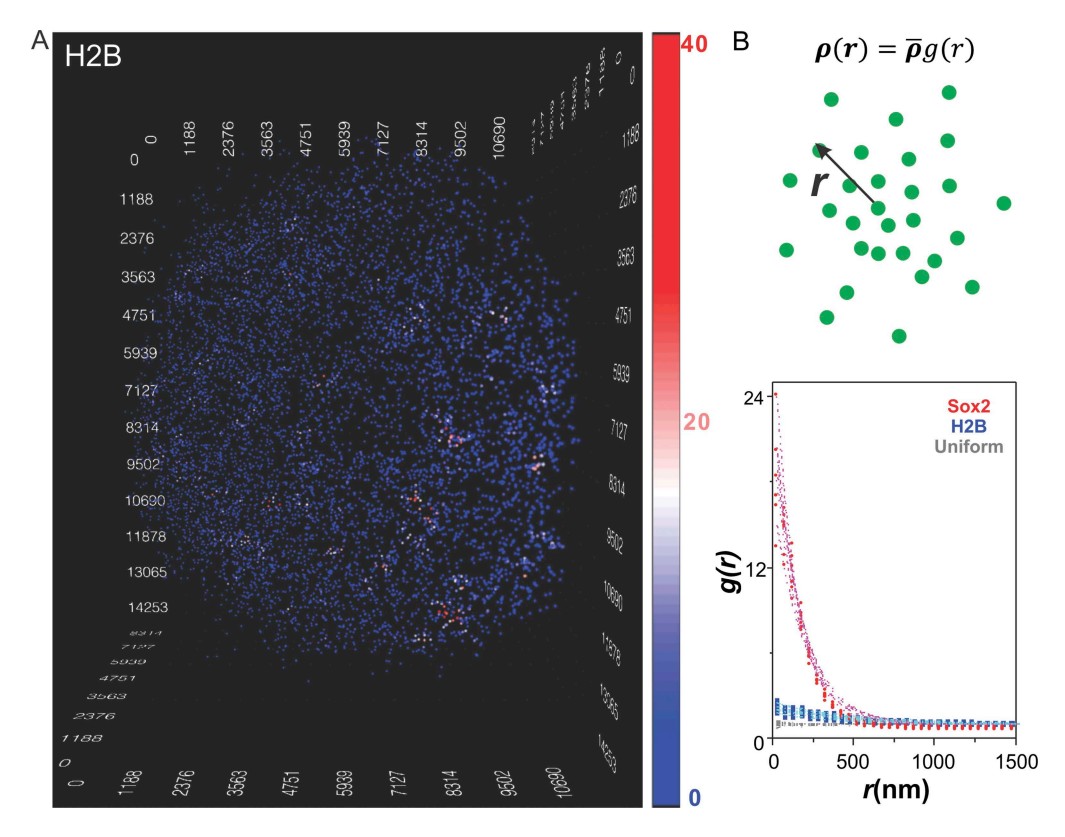

**Figure 2**. Clustering of Sox2 bound enhancers in the nucleus. (**A**) 3D density map of H2B distribution (n = 7000) in single ES cell nucleus. The imaging condition and analysis parameter set-ups were the same as HaloTag-Sox2 in *Figure 1*. The color map reflects the number of local neighbors that was calculated by using a radius of 400 nm. The unit of the x, y, z axes is nm. See *Video 4* for the full 3D rotation movie. (**B**) Upper: The pair correlation function *g(r)* measures the relative density of enhancer sites in a volume element at a separation r from single enhancer sites, given that the average density of enhancer sites in the whole volume is $\bar{\rho}$. See *Equations 3–7* for calculation details. Lower: Pair correlation function of Sox2 stable binding sites (red dots, n = 6), H2B (blue squares, n = 6), and simulated uniformly distributed particles (gray diamond, n = 5, *Video 5*) fitted with the fluctuation model (dotted lines) (See *Equations 10–13*). The obtained fluctuation amplitude and range for each curve are in *Supplementary file 1*.

The following figure supplements are available for figure 2:

**Figure supplement 1**. Quantification of clustering by pair correlation function.

**Figure supplement 2**. Temporal profiles of individual clusters and the number of localization detections per frame.

binding sites. Instead, we found the shorter-lived Sox2 binding sites showed greatly reduced fluctuation amplitudes of the pair correlation function curves (*Figure 2—figure supplement 1C–D*). We also note that in many cases, we observed little or no clustering of short-lived Sox2 binding sites within the same territories where longer-lived stable Sox2 binding site clusters can clearly be observed (*Videos 3 and 6*). These results suggest that the long-residence time filtering strategy that we deployed here likely enriches for specific binding site signals above the background of non-specific interactions consistent with what we observed previously (*Chen et al., 2014b*).

To further study the dynamic properties of EnCs, we used a time-counting analysis method (*Cisse et al., 2013*) to probe the temporal profiles of arrival times of stable binding events within individual clusters. Interestingly, we did not observe significant bursting behaviors as described for Pol II clusters (*Figure 2—figure supplement 2A–C*). These results are consistent with a model wherein Sox2 EnCs are relatively stable during the period (~20 min) of image acquisition. Because Sox2 bound

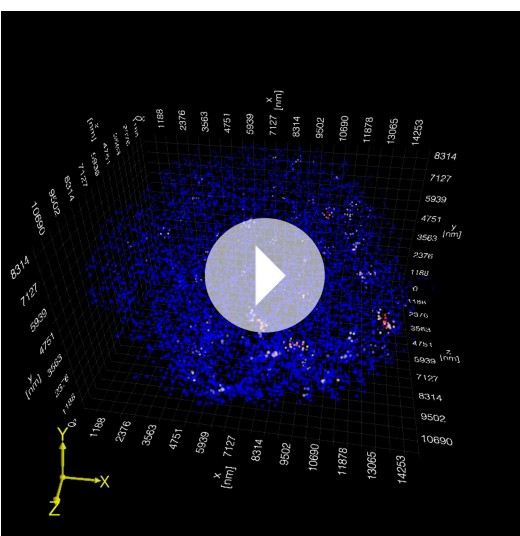

**Video 4**. Reconstructed H2B distribution in the live ES cell nucleus. HaloTag-H2B sites (7000) were localized, tracked, and reconstructed with a color map same as that of **Figure 2A**. The unit is nm.

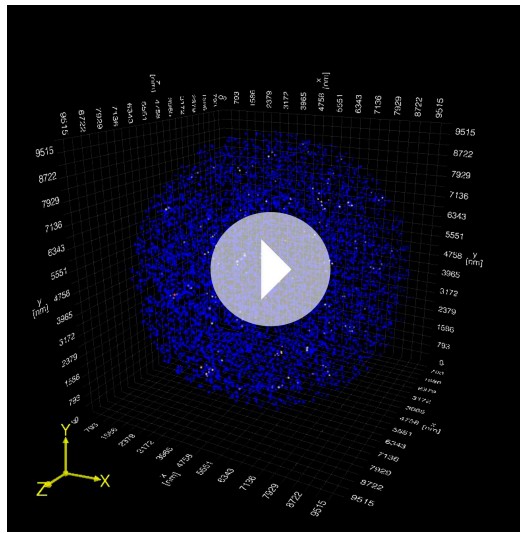

**Video 5**. Uniformly distributed, simulated positions in a nucleus. Uniformly distributed positions (7000) were presented with a color map same as that of **Figure 1C**. The unit is nm.

enhancers are chromatin based structures, we note that previous FRAP (Fluorescence recovery after photo-bleaching) experiments on core histone components (*Kimura and Cook, 2001*) found that large-scale chromatin structures in live cells appeared stable with a half-life of >2–4 hr which is much longer than the duration of our imaging experiments. These findings suggest that the enhancer clustering we observed here likely reflects the average 3D genome organization within reasonably short temporal length scales.

## Sox2-enhancer clusters are largely segregated from heterochromatin

It has long been proposed that the 3D space inside a cell nucleus is sub-divided into highly active gene enriched regions (so-called 'euchromatin') and largely inactive gene regions (i.e., 'heterochromatin'). To probe the spatial relationship between Sox2 EnCs and heterochromatic regions (HCs), we generated dual labeled ES cell lines that stably express HaloTag-Sox2 and GFP-HP1. HP1 protein is enriched in pericentromeric and peripheral HCs (*Grewal and Elgin, 2002*) that form non-diffraction limited structures in the nucleus (*Figure 3B,E*, *Figure 3—figure supplement 1C*, *Figure 3—figure supplement 2*, and *Videos 7,8,9*). To map the EnCs and HCs in the same cell, we first deployed a widefield, two-color imaging scheme (*Figure 3A*) in which we used a low-excitation, long-acquisition time imaging condition (2 Hz) to map Sox2 stable binding sites in the nucleus while we recorded the images of GFP-HP1 before and after the SMT experiment (*Video 7*). After localization and tracking of stable binding sites, we used a 2D kernel density estimator to generate an intensity map of EnCs in the nucleus (See 'Materials and methods' for details of image acquisition and registration; *Figure 3—figure supplement 1B*) and then superimposed the EnC intensity map with the HC map as two different color channels (*Figure 3B* and *Figure 3—figure supplement 1C*). We observed that EnCs and HCs are generally not co-localized spatially (*Figure 3B*). To gain a more quantitative measurement of these two distinct sub-nuclear regions, we tested the pixel-to-pixel correlation between EnC and HC intensity maps from individual cells. Pixels with high levels of EnC intensities generally showed low levels of HC intensities and vice versa (*Figure 3C*). The Pearson correlation test gave an averaged coefficient (Rho) of 0.11 ± 0.028 (n = 8) (*Figure 4—figure supplement 1B*), suggesting that the location of EnCs and HCs is indeed very weakly correlated in the nuclear volume of ES cells. We also used pair cross-correlation analysis (*Veatch et al., 2012*) to characterize the spatial relationship between EnC and HC regions (see *Equations 8–9* for details of calculation). Unlike autocorrelation which measures the degree of self-clustering, pair cross-correlation examines the degree of co-clustering and co-localization between two types of molecules. As expected, the EnC and HC were shown to be clustered as their self-cross (auto) correlation curves start with values

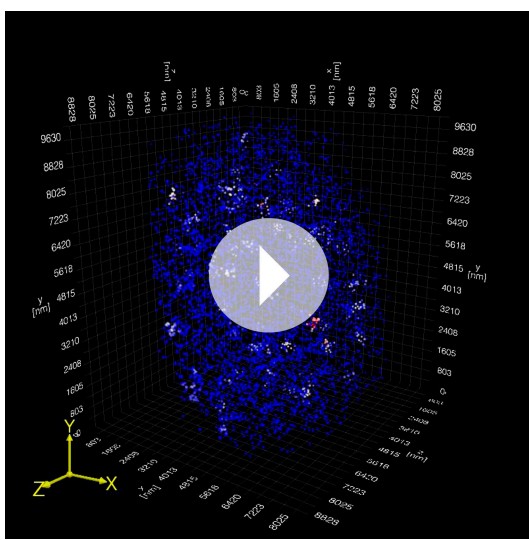

**Video 6**. Transient Sox2 binding sites in the live ES cell nucleus. HaloTag-Sox2 transient binding sites (7000, <3 s) were displayed with a color map same as *Figure 1C*. The unit is nm.

significantly above 1 and gradually converge to 1 with increased correlation radii (*Figure 3D*). By contrast, the pair cross-correlation function between EnC and HC intensity maps showed no apparent spatial correlation other than weak exclusion in the range of radii smaller than 600 nm (*Figure 3D*). To minimize the possibility that some bias may have been introduced by our 2D wide-field imaging and analysis pipeline, we also analyzed the spatial distributions of HCs and EnCs in single cells by lattice light-sheet imaging. Indeed, single molecule tracking confirmed and further strengthened the segregated relationship between HCs and EnCs in the 3D nucleus (*Figure 3E* and *Video 9*). Importantly, these results, taken together, suggest that Sox2 specific binding sites appear less frequently in HCs as most of the stable/specific binding sites were found to be outside of HCs. Consistent with this notion, we observed that levels of Sox2 in HCs are generally significantly lower than Sox2 levels in surrounding sub-nuclear regions, consistent with a reduced association of Sox2 to HCs (*Figure 3—figure supplement 2*).

## Sox2 EnCs overlap with a subset of pol II enriched regions

To investigate the spatial relationship and inferred functional correlation between Sox2 enhancers and RNA Pol II distribution in the nucleus, we generated an ES cell line stably expressing HaloTag-Sox2 and a Dendra2 tagged Rpb1 mutant that is resistant to α-amanitin (*Cisse et al., 2013*). These dual labeled ES cells were able to proliferate in the presence of α-amanitin, indicating that the tagged Rbp1 replaced the endogenous subunit in the RNA Pol II complex without interfering with its normal transcription function. To acquire super-resolution images of Pol II and Sox2 EnCs in the same cell, we first mapped Sox2 EnC clusters by deploying low-excitation and long-acquisition times (2 Hz) for detecting stable DNA bound JF646–HaloTag-Sox2 molecules. Next, we performed live-cell PALM experiments by photo-activating Dendra2 tagged Pol II molecules (See 'Materials and methods' for details of image acquisition and registration). The final reconstructed images are shown in *Figure 4A*. We also performed pair auto- and cross-correlation analysis with Pol II and EnC intensity maps (*Figure 4C*). Interestingly, results from autocorrelation analysis suggested that Pol II molecules are somewhat more evenly distributed in the nucleus than the highly clustered Sox2-enhancers. Specifically, Sox2 EnC autocorrelation curves generally start with higher values (higher packing densities) and more quickly converge to 1 with increased correlation radii (tighter packing) (*Figure 4C*) compared with Pol II autocorrelation curves. However, it is worth noting that we did detect significant and distinct local Pol II density fluctuations (*Figure 4A,C*) consistent with previous reports using other imaging modalities (*Cisse et al., 2013*; *Zhao et al., 2014*). To better quantify the spatial relationship between the distribution patterns of Pol II and Sox2 EnCs, we next determined the pixel-to-pixel correlation between EnC and Pol II intensity maps generated from individual cells (*Figure 4—figure supplement 1A*). The Pearson correlation test gave an average coefficient (Rho) of 0.44 ± 0.046 (Pol II high mask) (n = 8, average p-value from each test <4.36E-45), suggesting that unlike the relationship between EnCs and HCs, EnC regions are generally correlated with Pol II occupancy in ES cells. Pair cross-correlation function also suggested a significant degree of co-localization/clustering between Sox2 EnCs and Pol II–enriched regions as the cross-correlation curves start with values significantly greater than 1 and gradually converge to 1 (*Figure 4C*). However, we note that, due to the tighter clustering of Sox2 enhancers, most Sox2 EnC regions contained significant levels of Pol II whereas only a subset of Pol II enriched regions overlap with Sox2 EnCs (*Figure 4A–B*). The partial overlap between Sox2 EnCs and Pol II enriched regions is entirely consistent with previous genome-wide analysis showing that Sox2 only targets a subset of transcribed genes involved in maintaining ES cell

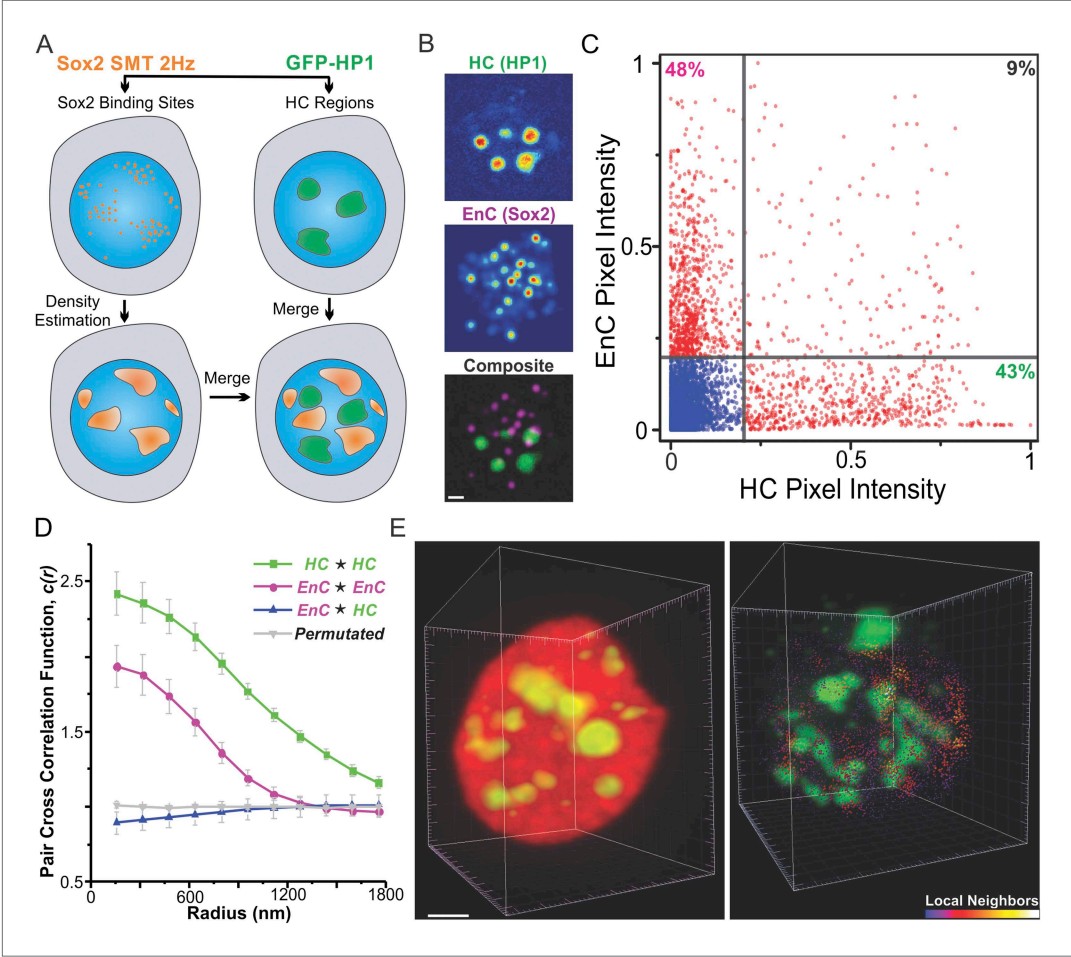

**Figure 3**. Sox2 enhancer clusters and heterochromatin regions are not co-localized. (**A**) Two color imaging to probe the spatial relationship between enhancer clusters and heterochromatin regions. Sox2 stable binding sites were mapped by low-excitation 2D single molecule imaging condition (**Video 7**). 2D kernel density estimator was used to generate the 2D intensity map of enhancer clusters in the nucleus (**Figure 3—figure supplement 1B**). The intensity map of heterochromatin regions was obtained by using the GFP-HP1 channel (**Figure 3—figure supplement 1A**). The composite image was constructed by merging the two intensity maps as two separate color channels. (**B**) Single-cell exemplary images of the HC, EnC intensity maps, and the composite. See **Figure 3—figure supplement 1C** for more examples. (**C**) The pixel-to-pixel intensity plot from the HC and EnC intensity maps shown in (**B**). The x, y value of each point is the intensity of HC (x) and that of EnC (y) from the same pixel. Pixels with low Sox2 EnC and HC intensity values were considered as background signals (blue points). The percentile of points in each quarter (over the total number of red points) was indicated in the corner of the region. (**D**) Pair auto- and cross-correlation function of HC (auto, green), EnC (auto, pink), HC ⋆ EnC (blue), and permutated (gray) images to investigate the spatial relationship between HC and EnC regions in single cells. ⋆, denotes the cross-correlation operator. See **Equations 8–9** for calculation details. Permutation was performed by randomizing pixels spatially within the nucleus mask for both HC and EnC images prior to calculating the cross-correlation function. (**E**) 3D spatial relationship between heterochromatin regions and Sox2 enhancer clusters determined by two color lattice light-sheet imaging. The HaloTag-Sox2 over-labeled image (left) shows fluorescent intensities contributed by all JF549-HaloTag-Sox2 molecules and the single-molecule tracked image (right) only shows the stable Sox2 binding site distribution. See **Videos 1, 8, and 9** for the exemplary raw data and the full rotation movie.

The following figure supplements are available for figure 3:

**Figure supplement 1**. Heterochromatin and Sox2 EnC spatial relationship.

**Figure supplement 2**. Probing Sox2 levels in heterochromatin regions.

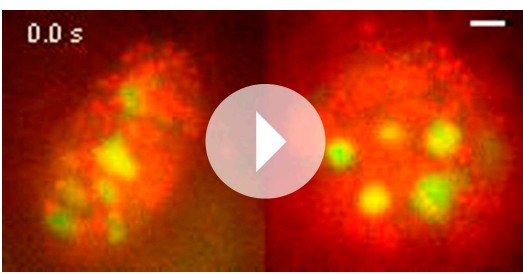

**Video 7**. Map stable Sox2 binding sites in GFP-HP1 labeled cells. Low excitation and long acquisition time (500 ms) wide-field imaging was used to map Sox2 stable binding sites in the GFP-HP1 labeled cells.

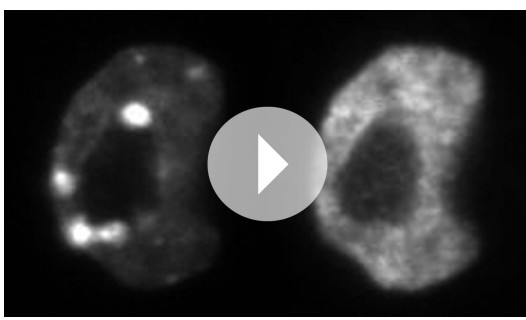

**Video 8**. Two color light-sheet imaging of Sox2 over-labeled GFP-HP1 ES cells. HaloTag-Sox2 is over labeled with JF549 ligand. Light-sheet imaging was performed with a z step of 200 nm.

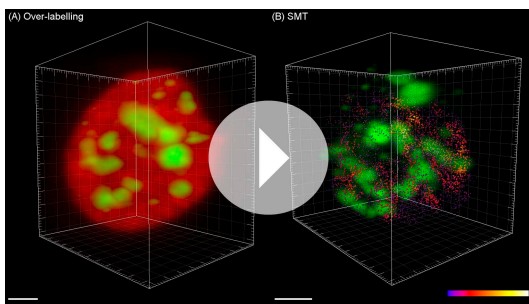

**Video 9**. 3D spatial relationship between heterochromatin and Sox2 enhancer clusters. (**A**) 3D reconstruction of over-labeled JF549 HaloTag-Sox2 and GFP-HP1 in single cell nucleus. (**B**) 3D reconstruction of JF549 HaloTag-Sox2 stable binding events (7000) (residence time >6 s) and GFP-HP1 in single cell nucleus. Scale bar, 2 μm. The color map reflects the number of local neighbors that was calculated by using a canopy radius of 400 nm.

identity (*Chen et al., 2008*). Many actively transcribed genes (including house-keeping genes) are likely subject to regulation by TFs other than Sox2 or Oct4. These results also suggest that Sox2 enhancer driven gene regulation is largely confined locally within distinct EnCs. Although not detectable in our assays, we assume that sub-nuclear regions outside Sox2 EnCs contain different actively transcribed cis-element clusters that also overlap with other Pol II enriched regions.

## Distinct Sox2 diffusion and binding behaviors in EnCs vs heterochromatin

In our recent study, we found that in ES cells, Sox2/Oct4 search for their target binding sites via a 3D diffusion dominant mechanism with an average dynamic 3D searching time ($\tau_{3D}$) of 3–4 s (*Chen et al., 2014b*). However, we were not able to determine whether Sox2 might actually behave differently in distinct sub-nuclear compartments and how enhancer clustering might influence the TF search process. In light of our new finding that the 3D space within the ES cell nucleus can be divided into distinct EnC and HC regions, it became possible to probe the behavior of Sox2 target search dynamics in different chromatin compartments. To address this important and functionally relevant question, we took advantage of recently developed HaloTag dyes (JF549 and JF646) for multiplexing SMT experiments (*Grimm et al., 2015*). We dual labeled HaloTag-Sox2 molecules in the same cells with JF549 and JF646 HaloTag-ligands (*Figure 5A*). Next, we mapped Sox2 EnC clusters by deploying low-excitation and long-acquisition times (2 Hz) for detecting stable bound JF646–HaloTag-Sox2 molecules. At the same time, we tracked the fast diffusing/ binding dynamics of JF549-HaloTag-Sox2 molecules by using high-excitation and short-acquisition times (100 Hz) (*Figure 5—figure supplement 1A* and *Video 10*). We also generated a binary mask for enhancer cluster regions and divided the Sox2 single-molecule tracks into in-mask fragments and out-mask fragments (See 'Materials and methods' for details). We note that tracking was performed without knowledge of the mask thus ensuring an unbiased track division. We calculated diffusion coefficients from in-mask track segments (n = 6 cells) (*Figure 5A*) and found that most of the molecules inside EnCs are in the bound state (64 ± 7.8%) and only ~36% are rapidly diffusing. These results suggest that Sox2

molecules in EnCs generally spend less time in diffusion before engaging with chromatin and thus have a shorter $\tau_{3D}$. Similarly, we investigated the fast diffusing/binding population of Sox2 within HCs using an analogous strategy (*Figure 5B* and *Figure 5—figure supplement 1B*, *Video 11*).

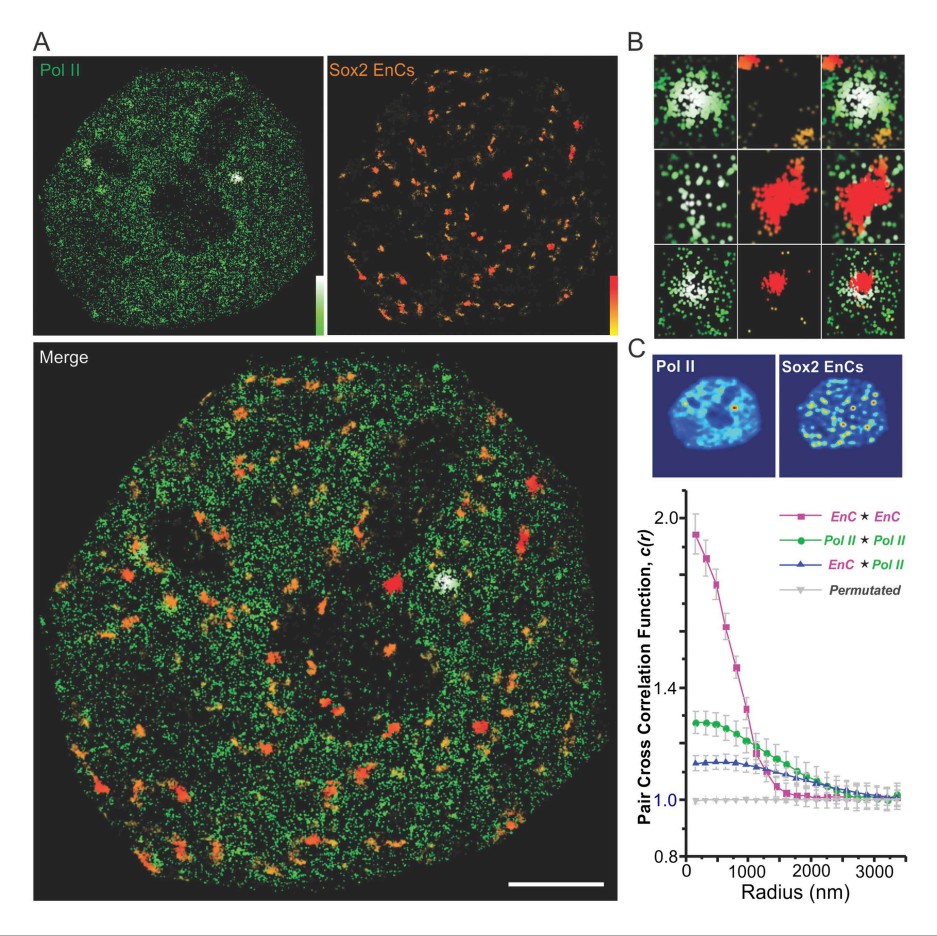

**Figure 4**. Sox2 targets a subset of Pol II-enriched regions in the nucleus. (**A**) Upper left: a live-cell 2D PALM super-resolution image of Dendra 2 Pol II. Upper Right: Sox2 enhancer clusters mapped by time-resolved, 2D single-molecule imaging/tracking. Stable binding events (>2 s) were shown. The color map that reflects number of local neighbors was displayed at the bottom right corner of each image. The canopy radius for calculation is 400 nm. Lower: the superimposed image of Pol II and Sox2 EnCs; Scale bar: 2 µm. (**B**) Selected zoomed-in views from (**A**); only a subset of Pol II enriched regions are targeted by Sox2. (**C**) Upper: single-cell exemplary images of the Pol II and EnC intensity maps calculated by 2D kernel density estimation. Lower: pair auto- and cross-correlation function of Pol II (auto, green), EnC (auto, pink), Pol II ⋆ EnC (blue), and permutated (gray) images to investigate the spatial relationship between Pol II enriched and EnC regions in single cells. ⋆, denotes the cross-correlation operator. See *Equations 8–9* for calculation details. Permutation was performed by randomizing pixels spatially within the nucleus mask for both Pol II and EnC images before calculating the cross-correlation function.

The following figure supplement is available for figure 4:

**Figure supplement 1**. Spatial correlation between Sox2 EnCs and Pol II enriched regions.

Consistent with our previous observations (*Figure 3*), stable DNA binding events/sites are distinctly low (16 ± 4.5%) in HCs, compared with their frequency in EnCs (64 ± 7.8%) and in whole nuclei (38 ± 4.3%) (*Figure 5A–C*). However, interestingly, we observed a significant population of Sox2 molecules (26 ± 8.4%) within HCs that diffuse with much slower rates (0.61 ± 0.13 µm²s⁻¹) than the average Sox2 diffusion rates (~2.7 ± 0.63 µm²s⁻¹) in whole nuclei (*Figure 5B–D*). This finding suggests that, in certain regions inside HCs, Sox2 diffuses slower. In good agreement with this observation, a previous report demonstrated via Fluorescence Correlation Spectroscopy (FCS) measurements that even GFP molecules diffuse much slower in heterochromatic regions possibly due to molecular crowding effects (*Bancaud et al., 2009*). We pooled and analyzed all the SMT tracks

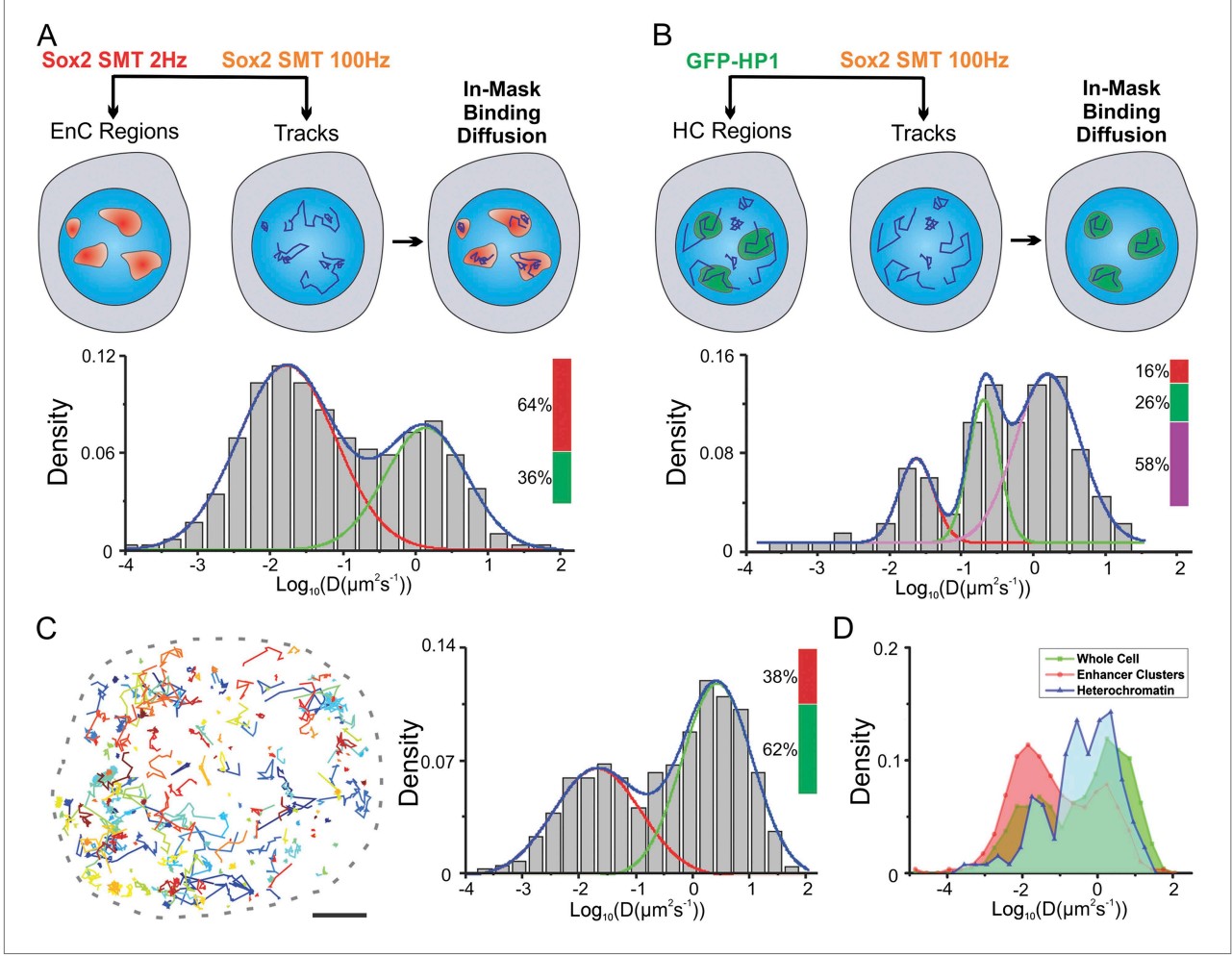

**Figure 5**. Two-color imaging reveals differential Sox2 behavior within enhancer clusters vs heterochromatin. (**A**) Two color single-molecule imaging to probe Sox2 binding and diffusion dynamics in enhancer clusters. EnC regions were first mapped by the low-excitation, long-acquisition time condition. Then, the diffusion coefficient histogram of tracks within the EnC regions was calculated and displayed in the lower panel (n = 6 cells). See *Figure 5—figure supplement 1A* and *Video 10* for more details. The obtained histogram was well fitted with two Gaussian peaks to a fast diffusion (green, D = 1.4 ± 0.18 μm²s⁻¹) and a bound (red, D = 0.017 ± 0.006 μm²s⁻¹) population. (**B**) Two color imaging to characterize Sox2 binding and diffusion dynamics in heterochromatin regions. Heterochromatin regions were first mapped by using the HP1-GFP marker. Then, the diffusion coefficient histogram of tracks within the heterochromatin regions was calculated and displayed in the lower panel (n = 9 cells). See *Figure 5—figure supplement 1B* and *Video 11* for more details. The histogram was well fitted with three Gaussian peaks to a fast diffusing (pink, D = 1.58 ± 0.25 μm²s⁻¹), a slow diffusion (green, D = 0.61 ± 0.13 μm²s⁻¹), and a bound (red, D = 0.023 ± 0.011 μm²s⁻¹) population. (**C**) Whole-cell Sox2 binding and diffusion dynamics. Single-molecule tracks were shown in the right panel. Data can be fitted by two Gaussian peaks to a fast diffusing (pink, D = 2.7 ± 0.63 μm²s⁻¹) and a bound (red, D = 0.021 ± 0.008 μm²s⁻¹) population (n = 12 cells). Scale bar: 2 μm. (**D**) Histograms from (**A**–**C**) were overlaid.

The following figure supplement is available for figure 5:

**Figure supplement 1**. Regional specific diffusion and binding dynamics.

obtained from single cells together and found that the majority of Sox2 molecules are in a diffusing mode (62 ± 4.3%, n = 12 cells) (*Figure 5C*), consistent with a 3D diffusion dominant search mechanism.

These results suggest that the Sox2 target search process is likely modulated by the spatial organization of enhancer clusters in the ES cell nucleus. Specifically, inside individual EnCs, Sox2 molecules appear to spend significantly more time binding to either naked DNA or chromatin with relatively short 3D diffusion periods. By contrast, Sox2 molecules that travel from one EnC to the next EnC navigate and tunnel through HCs by a 3D diffusion dominant long-range mode.

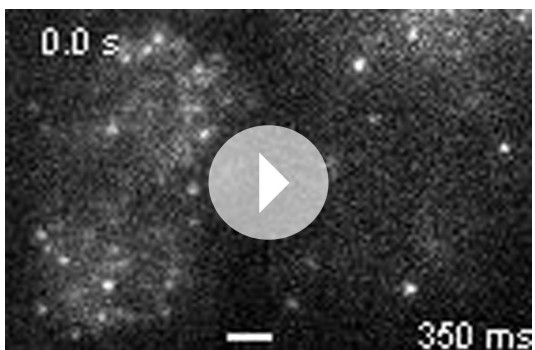

**Video 10**. Tracking Sox2 binding/diffusion dynamics within enhancer clusters. Two color single molecule imaging was performed with JF646 channel (Left) for mapping the enhancer cluster regions and JF549 (right) for tracking fast Sox2 diffusion/binding dynamics.

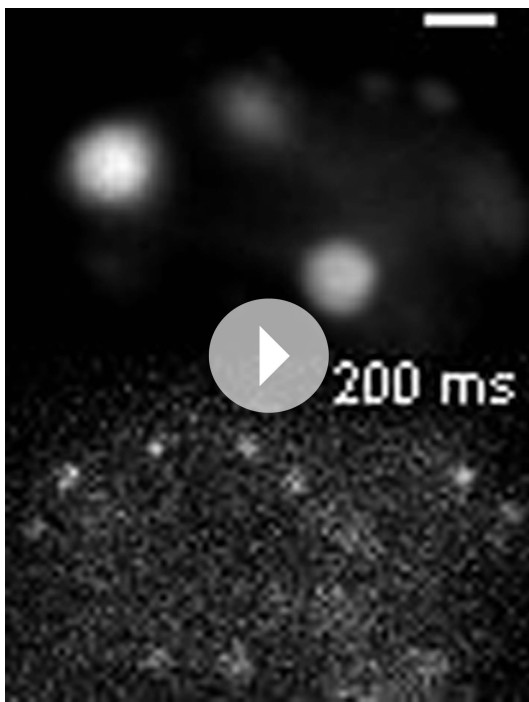

**Video 11**. Tracking Sox2 binding/diffusion dynamics in heterochromatin regions. Two color imaging was performed with the GFP channel (upper) for mapping the heterochromatin regions and JF549 (lower) for tracking fast Sox2 diffusion/binding dynamics.

## Enhancer clustering modulates global search efficiency and local rapid tuning

To further dissect the potential effects of enhancer clustering on TF target search dynamics, we investigated the search process carried out by Sox2 confronted with different degrees of enhancer clustering. The manipulations required to modulate enhancer clustering posed significant experimental challenges. Moreover, because of its probabilistic nature, the target search process cannot be adequately described by ordinary differential equations nor traditional binding kinetic equations, because they typically rely on mass reaction rates and assume that substrate concentrations are invariant across a large field of view. As discussed extensively in the literature (*Robert and Casella, 2005*), one of the most effective ways to dissect a random process based behavior is through computer-generated Monte Carlo algorithms that simulate the Brownian motion of TFs in a confined 3D sphere (cell nucleus) with multiple target traps (*Equations 14–15*, *Figure 6A*, *Video 12*, See 'Materials and methods' for parameter selection criteria. See *Figure 6—figure supplement 1A–C* for the validation of TF Brownian simulation). With such a set-up, we can arbitrarily manipulate the distribution of target sites in the nucleus, precisely control the initial TF injection position and then record the first 3D passage time ($\tau_{3D}$)—the duration from the initial injection to the point when the TF hits a target for the first time in the nucleus.

Having developed this simulation program (*Video 12*), we first tested how enhancer clustering would affect the global target search efficiency by injecting the TF randomly into nuclei with different degrees of enhancer clustering (*Figure 6A*). It is important to note that overlaps between targets were not allowed in our simulation experiments. Specifically, the minimal distance (80 nm) allowed between the centers of two targets is twice that of the target radius (40 nm). Interestingly, we found that it took increasingly longer $\tau_{3D}$ for TFs to reach their target as enhancer sites became more densely packed. This suggests that enhancer clustering may actually decrease global TF target search efficiency in the nucleus. In support of this result, other groups observed similar effects

of receptor clustering on ligand binding (*Goldstein and Wiegel, 1983*; *Care and Soula, 2011*). Specifically, the 'apparent' macroscopic ligand binding association rates decrease with increased densities of receptors within clusters while the microscopic rates remained the same. To further minimize the possibility that merged targets might create larger binding sites, we reduced the radius of targets to 30 nm while maintaining the minimal distance between the centers of two targets at 80 nm. In this case, there was no possibility of contact or merging between targets. Under these conditions, very similar simulation results were observed (*Figure 6—figure supplement 2A*). Thus, it seems unlikely

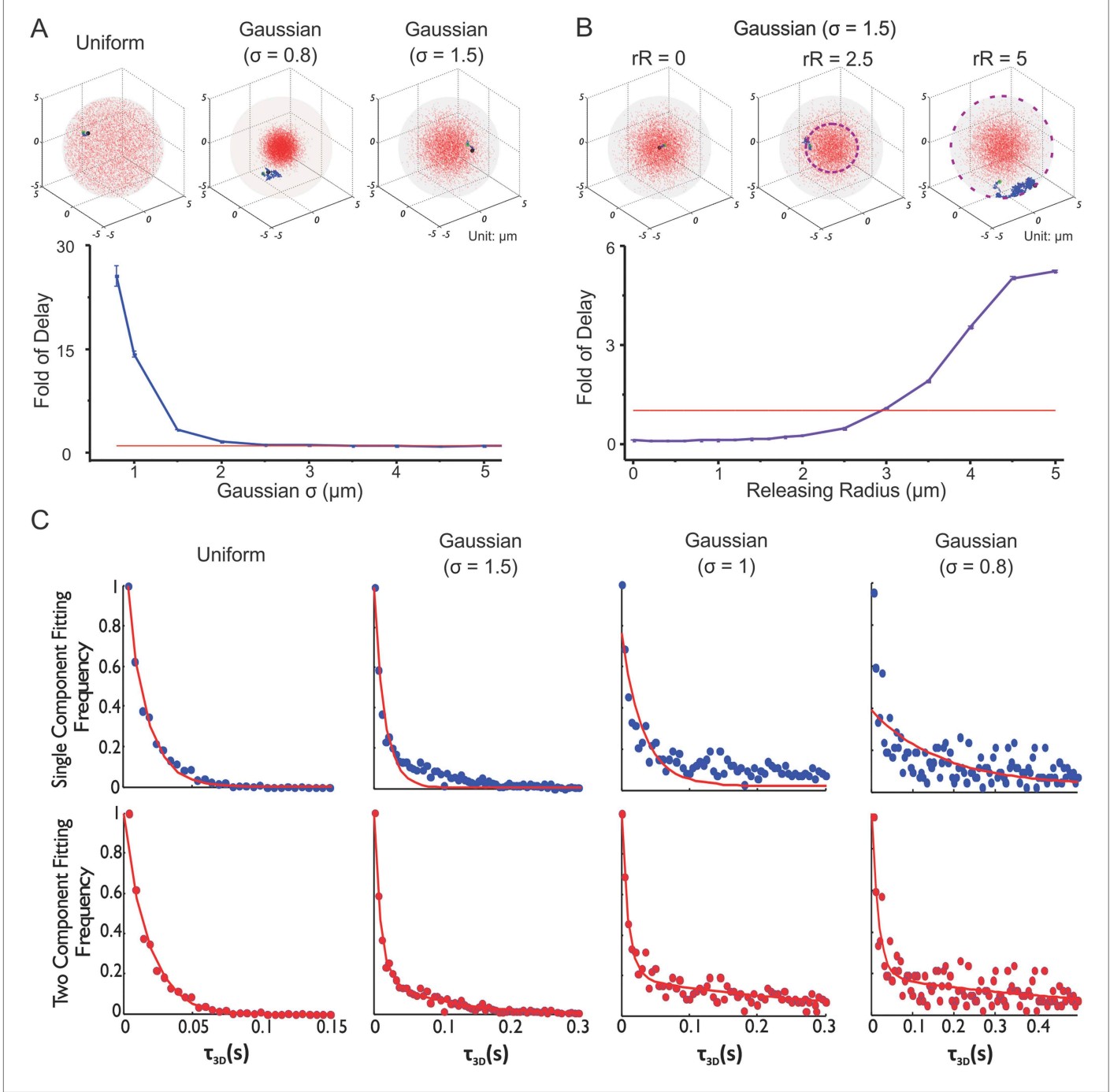

**Figure 6**. Enhancer clustering modulates global search efficiency and uncouples target search to a long-range and a local component. (**A**) Monte Carlo simulation of TF target search in the nucleus to test the effects of target site distribution on the first passage 3D time ($\tau_{3D}$). Fold of Delay is defined as the ratio of the average $\tau_{3D}$ in the clustered case to the average $\tau_{3D}$ in the uniform case. In this experiment, the TF injection site is randomly selected in the nucleus with no overlap with targets. The degree of clustering is tuned by changing the indicated S.D. of the Gaussian distribution. See 'Materials and methods' for detailed simulation parameters. TF target search simulation experiments were performed independently 100 times of total 10 repeats for assessing the standard deviation. The Fold of Delay was plotted as a function of S.D. (Sigma) in the lower panel. (**B**) Monte Carlo simulation of TF target search in the nucleus to test the effects of releasing Radius (Kaur et al.) on the first passage 3D time ($\tau_{3D}$). The injection site is randomly constrained in a shell with the indicated releasing radius relative to the center of the cluster. Fold of Delay is defined same as in (**A**). TF target search simulation experiments were performed independently 100 times of total 10 repeats for assessing the standard deviation. The Fold of Delay was plotted as a function of

*Figure 6. Continued on next page*

*Figure 6. Continued*
Releasing Radius (Kaur et al.) in the lower panel. (**C**) The histogram distribution of τ₃D for the indicated condition is fitted with both either the single-component (upper) or the two-component (lower) decay model (*Equations 17–18*). The experimental conditions were the same as (**A**).
The following figure supplements are available for figure 6:

**Figure supplement 1**. TF 3D Brownian motion simulation.

**Figure supplement 2**. Effects of number of clusters and distance-between-targets on TF target search.

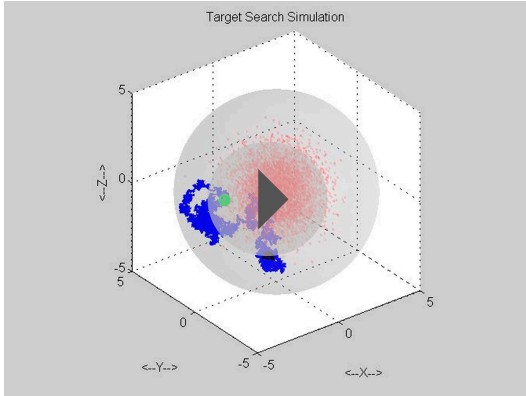

**Video 12**. TF target search simulation. An example of TF target search simulation in a single nucleus.

that the effects of target site clustering on τ₃D would be due to target site fusion. It is also important to note that, in our simulation experiments, the TF binding probability to target is 1. Since we defined the 'Fold of Delay' as τ₃D in the 'clustered' case normalized by τ₃D in the 'uniform' case, the binding probability (1 or not) should be identical under both uniform and cluster conditions. Consequently, the trends that we observe for 'Fold of Delay' should not alter significantly when the TF binding probabilities vary.

As expected, when we increased the number of clusters in the nucleus while holding the total number of sites constant, it took progressively shorter τ₃D for TFs to reach their target (*Figure 6—figure supplement 2B*). Under these conditions, we essentially increased the degree of randomness of enhancer distribution by dispersing the same amount of targets into more randomly localized clusters. We next probed the TF rebinding time in an individual cluster (*Figure 6B*). Specifically, we injected TFs at different radii of release relative to the center of an enhancer cluster. We found that τ₃D becomes reduced as the injection site approaches the center of the EnC (or when the local concentrations of enhancer sites increase). This result suggests that the TF target search dynamics is spatially modulated by enhancer density fluctuations in the nucleus such as we find in EnCs vs HCs (*Figure 5*). Specifically, the higher the local concentration of target sites, the shorter the time (τ₃D) it will take for a TF to reach a target site within an EnC. This relationship can also be verified mathematically by the Smoluchowski equations (*Equations 19–21*).

We next tried fitting the τ₃D histograms derived from different degrees of clustering to single or two-component decay models (*Equations 17–18*). Interestingly, a single component model failed to fit the data when the enhancer sites become more and more densely clustered while a two-component model fits the entire range of cluster density data well (*Figure 6C*). These results suggest that the enhancer clustering behavior itself may be sufficient to bifurcate the target search process into at least two components: a local search mode inside enhancer clusters and a long-range mode for searching outside of clusters. Together, these simulation results help clarify the effect of enhancer clustering on global TF target search efficiency in a non-equilibrium state and also reinforce the notion that the Sox2 target search process can follow two distinct modes: a local search process dominated by a binding dominant mechanism and a long-range mode for TFs to search between EnCs that is dominated by a 3D exploration mechanism as suggested previously by our independent SMT experiments (*Figure 5*).

## Epigenetic perturbations can disrupt Sox2-enhancer clustering and alter genome-wide binding profiles

As a first step towards deciphering the mechanisms that underlie enhancer clustering, we next asked whether modulation of the epigenome would change the Sox2 enhancer clustering behavior in single live cells. Specifically, we applied our single-molecule, light sheet imaging strategy to map Sox2-enhancer 3D organization in TSA treated ES cells (*Figure 2—figure supplement 1B* and *Video 13*). Interestingly, the pair correlation function of Sox2 EnCs in TSA treated cells showed profiles of significantly decreased

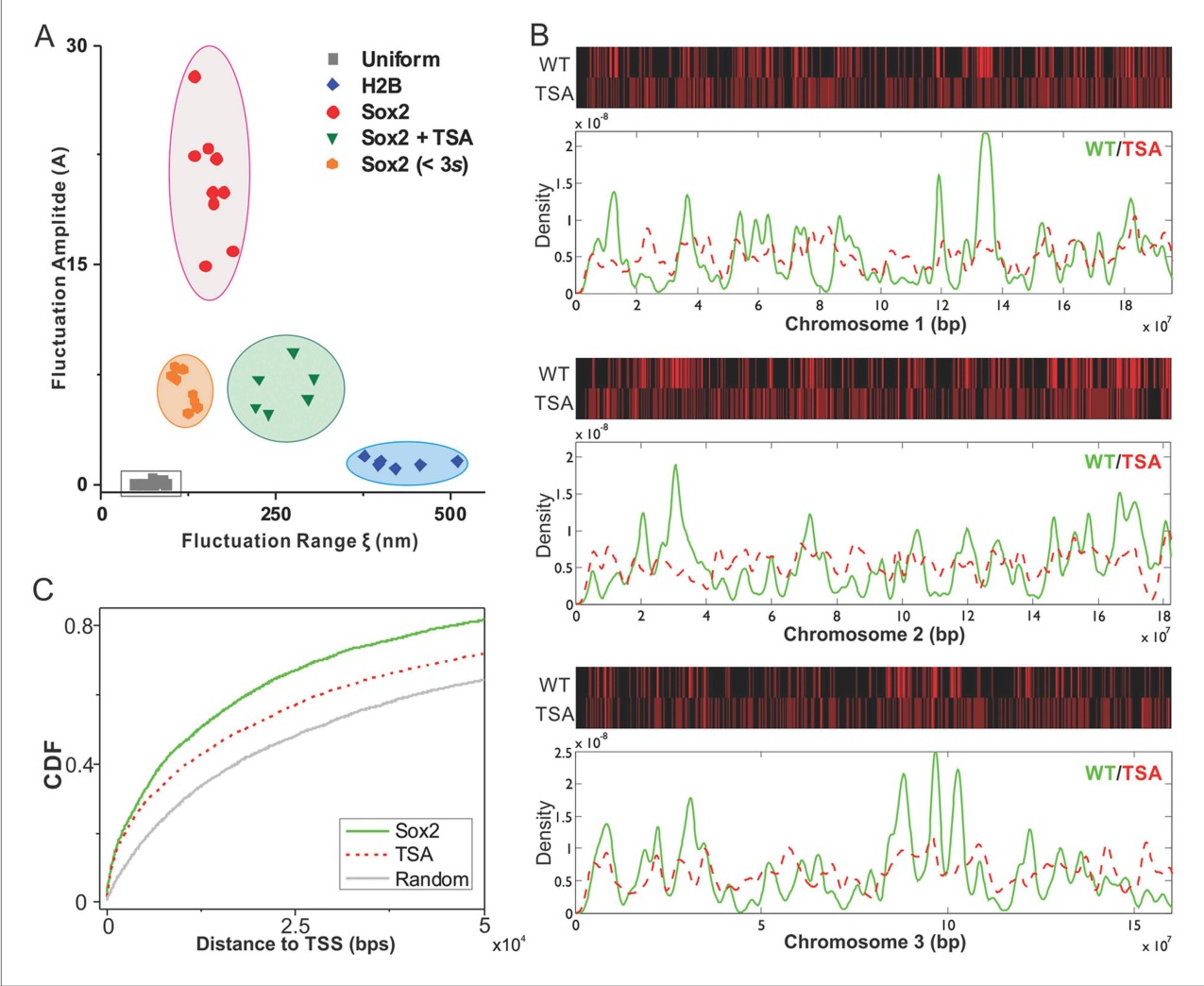

**Figure 7**. Epigenetic perturbation of enhancer clustering and genome-wide binding. (**A**) The fluctuation range (x) and amplitude (y) were obtained by fitting the pair-correlation function of the indicated dataset with the fluctuation model. *Figure 2* and *Figure 2—figure supplement 1*, *Equations 10–13*. *Supplementary file 1*. Data from the same condition were grouped in separate ellipses. (**B**) Sox2 ChIP-exo peak density distribution in the wild-type and TSA treated (red dotted) cells across chromosome 1, 2, 3. In the upper panels, each chromosome was divided to 500 bins. The color map correlates with the number of peaks in each bin. Top 7000 binding sites were considered in each condition. (**C**) Cumulative density histogram of the distances to transcription start sites (TSS's) of Sox2 ChIP-exo peaks in WT, Sox2 ChIP-exo peaks in the TSA treated cells (red dotted), and random genomic positions (gray).

clustering, more similar to those of H2B, indicated by the decreased fluctuation amplitudes and increased fluctuation ranges (*Figure 2—figure supplement 1D*, *Supplementary file 1* and *Figure 7A*). Thus, it seems that TSA treatment, thought to decondense chromatin, makes specific and stable Sox2 binding sites become more randomly distributed in the nucleus. One possibility is that dysregulation of histone deacetylation activities after TSA treatment significantly alters the Sox2 binding profile in the genome to a more random state; the other possibility is that TSA treatment redistributes the 3D localization of existing Sox2 binding sites in the nucleus. To distinguish between these two possible mechanisms, we performed Sox2 ChIP-exo experiments in TSA treated ES cells and compared the resulting Sox2 genome-wide binding profile to Sox2 chromosomal localizations in wild type (WT) ES cells. Upon TSA treatment, we observed a much more random distribution of Sox2 ChIP-exo peaks across different chromosomes and with regard to transcription start sites (*Figure 7B,C*, *Supplementary file 1*), favoring the scenario that TSA treatment significantly increased the chances for Sox2 to bind more randomly throughout the genome. These results suggest that a finely balanced epigenetic

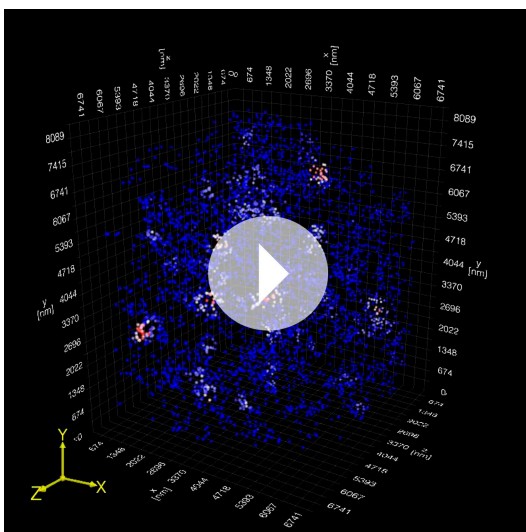

**Video 13**. Reconstructed Sox2 stable binding sites in the TSA treated live cell nucleus. HaloTag-Sox2 stable binding sites in the TSA treated live cell nucleus (7000, >3 s) were localized, tracked, and reconstructed with a color map same as that of **Figure 1C**. The unit is nm.

regulation can influence the maintenance of normal enhancer clustering in the nucleus.

## Discussion

A cornerstone of mechanisms regulating transcription is the productive encounter of one or more TFs with their cognate binding sites to form specific protein:DNA complexes that controls the transcriptional output of a gene. Despite more than 30 years of intense investigation, the dynamic TF target seeking process in live mammalian cells has remained poorly understood (**Halford, 2009**). Part of the problem is that, currently, there are few options available to directly investigate spatial enhancer organization in living cells. Here, we report a single-molecule, light-sheet imaging based strategy to reconstruct, in 3D, the Sox2-enhancer organization within live ES cells. We found that Sox2 enhancer sites form locally enriched clusters providing us an opportunity to tackle several fundamental questions. In particular, could the target search process be differentially regulated in distinct sub-nuclear regions such as enhancer clusters (EnCs) vs heterochromatic regions (HCs)? Using a suite of assays, we probed the transcription factor (TF) target search process and how the formation of enhancer clusters (EnCs) might modulate TF search parameters. We also determined the influence of spatially segregated and functionally distinct chromatin territories on transcription factor search modes within the mammalian nucleus. These findings support a model wherein gene regulation in eukaryotic cells operates in a manner dependent on the 3D spatial distribution of cis-elements that in turn influences differential target search features associated with local sub-nuclear environments (**Figure 8**).

### Integrating 3D enhancer spatial organization, target search dynamics, and localized transcription activity

Single-molecule tracking experiments coupled with in silico simulations reveal that enhancer clustering favors local spatial fine-tuning of search parameters at the expense of global search efficiency. In particular, we find that inside enhancer clusters, Sox2 displays significantly faster forward association rates (**Figures 5 and 6**), thereby increasing local TF concentrations, allowing rapid rebinding to stretches of open chromatin and probably also facilitating the local target acquisition process. The shortened $\tau_{3D}$ provides a greater opportunity for re-cycling pre-assembled TF complexes and taking advantage of cooperative interactions between TFs on chromatin. Interestingly, our simulation studies suggest that even subtle changes in the position of target genes within individual clusters can lead to alterations in local target search features. For example, gene targets at the center of EnCs can capitalize on different target search features relative to genes in the periphery of enhancer clusters (**Figure 8B**). These results suggest that the local TF target search mode may be exquisitely modulated within distinct sub-nuclear environments and serve as an important mechanism for fine-tuning the rates of TF complex assembly at specific cis-regulatory elements.

Two-color imaging revealed that enhancer clusters are spatially segregated from heterochromatic regions but overlap with a subset of Pol II enriched clusters (**Figure 4**). Single molecule tracking of Sox2 binding and diffusion dynamics in EnCs vs HCs indicates that in contrast to the previously estimated fraction (~74%) of Sox2 molecules engaged, on average, in 3D diffusion, the majority (~64%) of Sox2 molecules within EnCs were found to be in a chromatin bound state (**Figure 5**). This new finding suggests that local higher concentrations of 'open' chromatin in EnCs likely give rise to a dramatically reduced $\tau_{3D}$ leading the target search process to switch from a 3D diffusion dominant mode to a binding dominant search mechanism perhaps more similar to the action of LacI in bacteria (**Elf et al., 2007**). By contrast, we found a relatively low Sox2 bound fraction in heterochromatic regions.

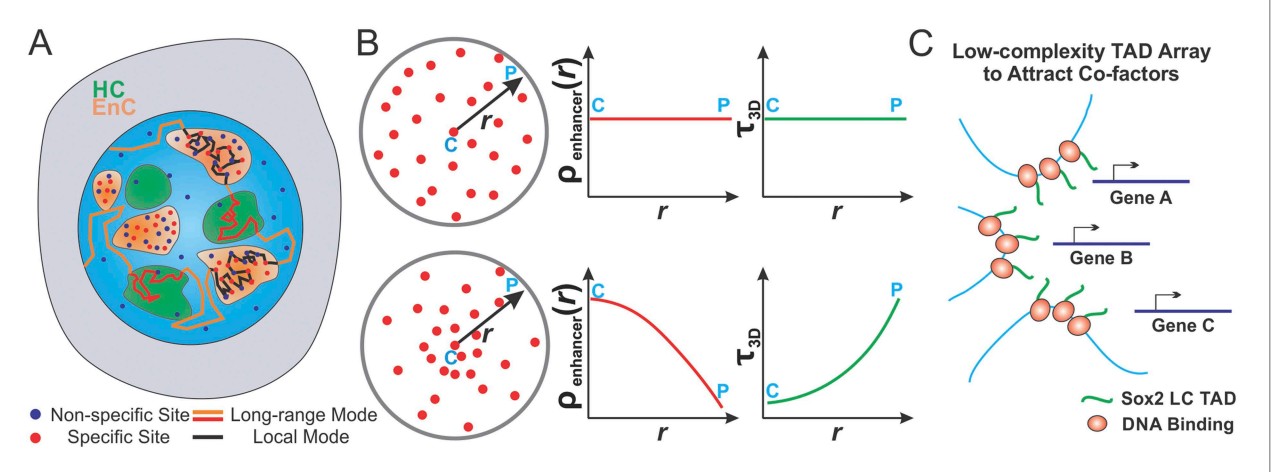

**Figure 8**. Spatially modulated target search and gene regulation in ES cells. (**A**) Sox2 stable binding sites form enhancer clusters that are segregated from heterochromatin regions. Sox2 searches for targets via a 3D diffusion dominant mode traveling between clusters and tunneling through hetero-chromatin regions. Inside individual enhancer clusters, Sox2 3D diffusion times were dramatically shortened due to the high concentrations of specific target sites, nonspecific open DNA or protein binding partners as indicated by *Figure 5A*. Thus, Sox2 target search is dominated by binding processes on chromatin. (**B**) Enhancer clustering modulates TF target search parameters and makes spatially controlled gene regulation possible by creating variation of local enhancer concentrations. Upper, uniform distribution of target sites generates invariant enhancer site density in the nucleus and TF would have the same average 3D times across the nucleus. Lower, target site clustering causes variations of local enhancer density which would affect local target search parameters (*Equations 19–21*). Genes at C position would be regulated differently compared to genes at P position. C and P stand for the Center and the Peripheral, respectively. (**C**) Enhancer clustering promotes the formation of local Sox2 Low Complexity (LC) transactivation domain (TAD) polymer arrays which could serve as multivalent platforms to dynamically recruit co-factors for localized chromatin modulation and gene activation.

Intriguingly, in certain HC regions, Sox2 diffuses with slower rates compared to the average rate in the nucleus. This result is consistent with a previous report that even GFP molecules diffuse much slower in HCs possibly due to molecular crowding effects (*Bancaud et al., 2009*). Our findings suggest that the Sox2 target search process can be divided into at least two distinct modes in the ES cell nucleus (*Figure 8A*), (1) a chromatin binding dominant, local mode within individual EnCs possibly involving Sox2 sliding along short stretches of naked DNA as demonstrated by our in vitro TIRF single-molecule experiments (*Chen et al., 2014b*) and (2) a 3D diffusion dominant long-range mode in which Sox2 molecules must tunnel through HCs and travel between EnCs. Our simulations also suggest that enhancer clustering itself is sufficient to generate these two modes of target search. Together, these findings reveal previously unappreciated principles governing Sox2 target search patterns within distinct sub-nuclear regions of ES cells and provide insights into how enhancer clustering can modulate local target search dynamics that could ultimately influence local transcription rates.

From the earliest cloning and characterization of classical sequence specific transcription factors, a striking yet puzzling feature was the discovery of simple repetitive largely unstructured amino acid motifs (i.e., Gln-rich, Pro-rich, acidic repeats) that serve as 'activation domains' (ADs) coupled to DNA binding domains (*Courey and Tjian, 1988*). More recent evidence suggests that such simple repetitive amino acid motifs, now referred to as low-complexity (LC) sequences, are found in a variety of regulatory proteins (such as FUS, TAF15, and EWS) and can be induced to form fibrous polymers in vitro to mediate interactions with the CTD of RNA polymerase in a phosphorylation-state dependent manner (*Kwon et al., 2013*). However, in vitro polymer formation required protein concentrations (0.7–2 mM), ~1000× greater than typical concentrations of TFs (low micro-molar) found in vivo (*Chen et al., 2014b*). One mechanism proposed to enhance polymer formation involves RNA molecules seeding higher-order assemblies via the intrinsic RNA binding capacity of select regulatory proteins (*Schwartz et al., 2013*). Interestingly, the activation domains of Sox2 are predicted to be unstructured LC domains enriched for G/S/P residues. Importantly, the C-terminal Sox2 AD contains five repeats of degenerate (G/S/D/H) Y (G/S/D/H) sequences that have been reported to form fibrous polymers in vitro (*Kwon et al., 2013*). We speculate that the in vivo Sox2-enhancer clustering observed in our live cell studies opens the possibility that local higher concentrations of both TFs and specific DNA binding

sites within EnCs may promote the formation of Sox2 LC AD containing polymers at least transiently. We envision that these Sox2-enhancer clusters could serve as multivalent docking sites for dynamic TF recruitment via weak protein:protein interactions potentially directed by LC containing proteins. Such 'clouds' of weak multivalent protein:protein interactions would be assisted by stronger sequence specific protein:DNA transactions that together build an activated enhanceosome. These transiently formed EnC clusters may, in turn, regulate local TF concentrations and dictate local target search dynamics of key transcriptional pre-initiation components including Pol II, GTFs, and chromatin remodeling complexes. It is tempting to speculate that the Sox2 enhancer cluster and its associated co-factors could thus form the local hub for coordinated and synergistic gene regulation (*Figure 8C*). Whether the EnCs we observe represent actively transcribed regions remains unclear but the significant co-localization between EnCs and Pol II would be consistent with such an interpretation. Since interactions between classical LC activation domains such as those reported in the original studies of Sp1-Sp1 and Sp1-TAF4 interactions have also been suggested to be important for DNA loop formation and transcription activation in vitro (*Mastrangelo et al., 1991*; *Su et al., 1991*; *Freiman and Tjian, 2002*), it will be instructive in the future to probe whether these prevalent LC mediated transactions also contribute to the maintenance and structural integrity of enhancer clusters.

Another intriguing feature of this revised model of gene regulation is that physical proximity but not direct or stable interactions between distal enhancer elements and gene proximal promoters is necessary for delivering transcription activation by cis-elements at a distance. We envision that enhancer clustering with its higher local TF/cofactor concentrations accompanied by altered target search features may be sufficient to serve as an alternative mechanism for achieving distal enhancer directed transcription activation long recognized as a hallmark of mammalian gene control. Our results also suggest that gene and promoter positioning in relationship to EnC and HC territories is critical for optimal fine-tuning of transcriptional activities. Important questions left unresolved by our present study include: how many genes/promoters are present within a cluster; what is the relationship between our 3D clusters and TADs (topologically associated domains); and are the enhancer binding sites within a cluster all from a given chromosome or is there evidence of transvection occurring as well?

## Linking 3D spatial distribution to linear genomic TF-binding profile

The enhancer clustering behavior that we observed fits generally with the concepts deduced from studies of linearly arrayed enhancers in the genome identified by ChIP-seq analysis (*Whyte et al., 2013*) and topological domains identified by Hi-C experiments (*Dixon et al., 2012*). These studies, taken in aggregate, suggest that gene transcription is compartmentalized within topological or spatial territories in the nucleus segregated from other silent regions. However, given the orthogonal nature of these diverse methods of probing genome organization, it is difficult at this stage to firmly establish either direct structural or functional links between the genome-wide ensemble studies and our observation of enhancer clustering by single molecule imaging. We note, for example, that enhancer clustering does not appear simply to result from differential chromatin packaging. Specifically, the fluctuations of Sox2 enhancer densities in the nucleus are smaller in sizes ($\varepsilon$) but much larger in amplitudes (A) than chromatin (H2B) densities (*Figure 7A*). Two distinct mechanisms could account for this observation, (1) Sox2 binding sites are already clustered in the linear genome and chromosome folding based on the polymer model (reviewed in *Tark-Dame et al., 2011* and *Fudenberg and Mirny, 2012*), automatically leads to 3D cluster formation even without TF directed chromatin looping interactions; (2) extensive active and TF directed long distance chromatin looping brings distal Sox2-enhancer sites along the linear chromosome to form local 3D clusters. To distinguish between these two potential scenarios, we analyzed the linear arrangement of Sox2 binding sites genome wide using ChIP-exo. Indeed as suspected, many Sox2 binding sites already form clustered arrays along the linear chromosome (*Whyte et al., 2013*). Interestingly, disrupting the epigenome alters the accessibility of linearly arranged Sox2 binding sites as well as the extent of 3D enhancer clustering (*Figure 7*). This finding suggests that linearly arrayed Sox2 binding sites likely contribute substantially to the formation of enhancer clusters in 3D but do not exclude the possibility that TF directed chromatin looping also contributes to such an organization of actively transcribed loci. Future modeling and simulation experiments will be required to functionally link the linear genomic localization of different TF binding sites with their 3D spatial distributions in the nucleus to gain further insights into genome organization and chromatin folding. It also remains unclear what forces or exogenous structures are in play to maintain the star-burst arrangement of enhancers in the nuclear volume.

## Concluding Remarks

Our studies provide a basis for understanding how the 3D organization of enhancers into localized clusters could affect TF target search dynamics and influence local transcription rates. Our imaging analysis of live ES cells suggests that the nucleus is partitioned into multiple levels of spatially segregated functional domains. For example, many Pol II enriched regions do not overlap with Sox2 EnCs although, as might be expected, most Sox2 EnCs do overlap with Pol II clusters (*Figure 4*). We also observed extensive residual 'dark' spaces that are not significantly occupied by either heterochromatin or Sox2 binding sites (*Figure 3B* and *Video 9*). It seems likely that other uncharacterized enhancer bearing sub-nuclear domains occupy these 'dark' territories and influence local gene activity not detected by our current assays. We speculate that galaxies of such 3D clusters of cis-regulatory domains are formed by specific binding of different combinations of TFs we have long suspected but could not discern from classical bulk biochemistry. It will be interesting in the future to complete this 3D mapping of the nucleome to discern which other cadre of factors might reside in these 'dark' regions. Another aspect to address will be the degree of spatial overlap between different functional regions created by the stable binding and local concentrations of different classes of TFs. Ultimately we would like to have a more complete understanding of how the 3D organization of these cis-elements specifically influences gene activity and what gene products and mechanisms underlie the formation of these clusters. Addressing these questions will be essential for a deeper understanding of how enhancer-mediated gene regulation works. The ongoing development of simultaneous multi-color super-resolution imaging systems, enhanced dye chemistry, and single gene locus labeling strategies will be essential to address these fundamental questions.

# Materials and methods

## ES cell culture

Mouse D3 (ATCC, USA) ES cells were cultured on 0.1% gelatin coated plates in the absence of feeder cells. The ES cell medium was prepared by supplementing knockout DMEM (Invitrogen, Carlsbad, CA) with 15% FBS, 1 mM glutamax, 0.1 mM nonessential amino acids, 1 mM sodium pyruvate, 0.1 mM 2-mercaptoethanol, and 1000 units of LIF (Millipore, USA). 1 day before imaging experiment, cells were plated onto a clean cover glass pre-coated with Matrigel (BD Biosciences, USA, 356230). In the TSA perturbation experiments, ES cells were treated with 50 nM TSA (Sigma-Aldrich, USA: T8552) for 8 hr prior to imaging and ChIP-exo mapping.

## Plasmid construction

Mouse HP1 (Cbx5 gene: NM_007626) cDNA was first amplified by PCR from ES cell cDNA libraries and then inserted into a custom-constructed Piggybac transposon vector that harbors the E1F alpha promoter, the internal ribosome entry site (IRES), and the PuroR gene. eGFP cDNA was further cloned to fuse with HP1 at its N-terminus.

## Stable cell line generation

Stable cell lines were generated by co-transfection of stable HaloTag-Sox2 ES cells established in our previous work (*Chen et al., 2014b*) with the HP1 overexpression piggybac vector and a helper plasmid that over-expresses Piggybac transposase (Supper Piggybac Transposase, System Biosciences, USA). 48 hr post-transfection, cells were subjected to puromycim (Invitrogen Carlsbad, CA) selection (1 µg/ml). After 3 days of selection, cells were maintained in their culturing medium with a 0.5 µg/ml final concentration of puromycin. Similarly, Dendra2-Rpb1 mutant cDNA was cloned into the piggybac vector and co-transfected into the HaloTag-Sox2 ES cells with the helper plasmid. α-amanitin (Sigma-Aldrich, USA: A2263) selection was conducted by using a final concentration of 3 µg/ml. 10 days after selection, stable cell clones appear on the place. For the long-term maintenance, 1 µg/ml α-amanitin was supplemented into the culturing medium. For electroporation, ES cells were first dissociated by trypsin into single cells. Transfection was conducted by using the Nucleofector Kits for Mouse Embryonic Stem Cells (Lonza, USA).

## Cell labeling strategy and preparation for imaging

All imaging experiments were performed in the ES cell imaging medium, which was prepared by supplementing FluoroBrite medium (Invitrogen, Carlsbad, CA) with 10% FBS, 1 mM glutamax, 0.1 mM

nonessential amino acids, 1 mM sodium pyruvate, 10 mM Hepes (pH 7.2–7.5), 0.1 mM 2-mercaptoethanol, and 1000 units of LIF (Millipore, USA).

For 3D lattice light-sheet imaging condition, we optimized HaloTag-JF549 concentrations in the medium to a final concentration of ~0.1 fM. The ligand molecules gradually diffuse into the cell and label the HaloTag-Sox2 molecules. Optimal single-molecule labeling density was achieved when the labeling rates equilibrated with the photo-bleaching rates. Due to the light-sheet selective plane illumination, the relative long acquisition time (40 ms), and ultralow ligand concentration in the medium, negligible fluorescent background signals were observed.

For 2D wide-field imaging condition, we first tested the optimal HaloTag-JF549 and HaloTag-JF646 labeling concentrations. Briefly, several concentrations of HaloTag-JF549 and JF646 (0.5 nM, 1 nM, 2 nM, and 5 nM) were used to treat cells for 15 min and then cells were washed with imaging medium for three times. The cover glasses were then transferred to live-cell culturing metal holders and mounted onto the microscope one by one. Proper HaloTag-JF549 or HaloTag-JF646 labeling concentrations were determined by the criterion that single-molecules can be easily detected under 2D imaging mode after a minimal 2–5 s pre-bleaching. After fixing the labeling concentration for each cell line, we then proceeded to perform the 2D single-molecule imaging experiments.

## Single-molecule imaging by lattice light-sheet microscope

3D single-molecule tracking experiments were performed via lattice light sheet plane illumination microscopy using a modified version of the multi-Bessel microscope described previously (*Gao et al., 2012*). The modification consists of a massively parallel array of coherently interfering beams comprising a non-diffracting 2D optical lattice, rather than a set of seven noninterfering Bessel beams. This creates a coherent structured light sheet that can be dithered to create uniform excitation in a 400 nm thick plane across the entire field of view. The experimental hardware is the same as before, except that a binary spatial light modulator (SXGA-3DM, Forth Dimension Displays, Valencia, CA) is placed conjugate to the sample plane, and a binarized version of the desired structured pattern at the sample is projected on the display. For imaging, a 500 mW cw488 nm (Coherent, Santa Clara, CA) or a 500 mW cw561 laser (MPB Lasertech, Edmonton, AB) were used. A custom 0.65 NA objective for excitation (Special Optics, Wharton, NJ) and a 25×, 1.1 NA objective for detection (Nikon, USA, MRD77220) are employed. A multi-band pass filter (Semrock, FF01-446/523/600/677-25) is placed before a CMOS camera (ORCA-flash4.0, Hamamatsu, Japan) to filter the excitation wavelengths. Single molecule imaging of individual cells was performed by serially scanning the entire cell nucleus through the light sheet at 20–50 ms exposure per 2D image and 300 nm z-steps resulting in a 3D imaging rate of 3 s per volume. Although significantly faster imaging rates are possible, these conditions were chosen to minimize photo-bleaching and phototoxicity, while specifically selecting stably bound (>3 s) molecules. Correlation of stable binding sites with heterochromatin regions was performed by first acquiring a single 3D volume of GFP-HP1 followed by single molecule imaging as described above.

## 3D PSF model, 3D single-molecule localization and image registration

3D localization (x, y, z) was conducted using FISH-QUANT software (*Mueller et al., 2013*). The PSF model can be described by the following equation:

$$I\left(x, y, z\right) = (A_0 e^{\frac{-(x-x_0)^2}{2\sigma_{xy}^2}} e^{\frac{-(y-y_0)^2}{2\sigma_{xy}^2}} e^{\frac{-(z-z_0)^2}{2\sigma_z^2}})_{PSF} + B,$$

(1)

where $A_0$ is the signal amplitude; $\sigma$ is the Standard Deviation (S.D.) of the Gaussian fit in the indicated direction, in our case S.D. of the x, y direction is the same; B is the number of background photon count.

Image registration and drift correction were performed by calculating the centroid displacement of total localization events from every 50 time points (2.5 min) and the resulting transformation matrix over time was applied to the data accordingly. We found that this method can efficiently correct drifts which were not significant (0–800 nm per minutes) within the correction time window. Any significantly drifted dataset was not used for later tracking analysis.

## Estimation of localization uncertainty

Localization uncertainty can be calculated by the estimator below (*Rieger and Stallinga, 2014*).

$$\Delta^2 = \frac{\sigma^2 + a^2/12}{N}\left(\frac{16}{9} + 4\tau\right) \tag{2}$$

With τ roughly equal to the ratio between the background intensity and the peak signal intensity, which can be directly obtained from the FISH-quant localization program. *a*, the voxel size in the selected direction. *N*, total photo count was calculated by integrating voxel photon counts covered by each Gaussian spot.

### 3D single-molecule tracking and 3D enhancer map reconstruction

U-track algorithm (*Jaqaman et al., 2008*) was used for 3D single particle tracking. For mapping Sox2 stable binding site in live cells, we only reconstructed the first events of track fragments which have step and end-to-end displacements less than 50 nm and have lengths longer than the indicated cutoff time. The final 3D image representation was performed by either ViSP (*El Beheiry and Dahan, 2013*) or Imaris.

### 2D single-molecule imaging

2D single molecule experiments were conducted on a Nikon Eclipse Ti microscope equipped with a 100× oil-immersion objective lens (Nikon, N.A. = 1.4), a lumencor light source, two filter wheels (Lambda 10-3, Sutter Instrument, Novato, CA), perfect focusing systems, and EMCCD (iXon3, Andor, UK). Proper emission filters (Semrock, Rochester, NY) was switched in front of the cameras for GFP, JF549, or JF646 emission and a band mirror (405/488/561/633 BrightLine quad-band bandpass filter, Semrock, Rochester, NY) was used to reflect the laser into the objective. For two color single-molecule experiments with JF646 and JF594 labeled HaloTag-Sox2, we used a 630-nm laser (Vortran Laser Technology, Inc.) of excitation intensity ~60 W cm$^{-2}$ and a 561-nm laser (MPB Lasertech, Edmonton, AB) of excitation intensity ~800 W cm$^{-2}$ and the acquisition times are 500 ms (630 nm) and 10 ms (561 nm). For two color experiments mapping the spatial relationship of heterochromatin and enhancer clusters, we used a SOLA light engine (Lumencor, Beaverton, OR) and a 561-nm laser (MPB Lasertech, Edmonton, AB) of excitation intensity ~50 W cm$^{-2}$ and the acquisition times are 100 ms (GFP) and 500 ms (561 nm).

After mapping stable Sox2 binding sites by using the JF646 dye, Dendra2-Rpb1 PALM experiment was performed using the 560-nm laser (MPB Lasertech, Edmonton, AB) of excitation intensity ~1000 W cm$^{-2}$ for single-molecule detection and a 405-nm laser (Coherent, Santa Clara, CA) of excitation intensity of 40 W cm$^{-2}$ for photo-switching of Dendra2-Rpb1. The acquisition time is 30 ms. Total ~10,000 frames were recorded. ~20k localized events were used for the final imaging reconstruction.

For two color experiments probing the Sox2 diffusion properties in heterochromatin regions, we used a SOLA light engine (Lumencor, Beaverton, OR) and a 561-nm laser (MPB Lasertech, Edmonton, AB) of excitation intensity ~800 W cm$^{-2}$ and the acquisition times are 100 ms (GFP) and 10 ms (561 nm). The microscopy, lasers, the SOLA light engine, and the cameras were controlled through NIS-Elements (Nikon, USA).

### 2D single-molecule localization and tracking

For 2D single molecule tracking, the spot localization (x, y) was obtained through 2D Gaussian fitting based on MTT algorithms (*Serge et al., 2008*) using home-built Matlab program. The localization and tracking parameters in SPT experiments are listed in the *Supplementary file 1*.

To map stable bound sites in the low excitation, slow acquisition (500 ms) condition, 0.05 μm$^2$/s was set as maximum diffusion coefficient (D$_{max}$) for the tracking. The D$_{max}$ works as a limit constraining the maximum distance (r$_{max}$) between two frames for a particle random diffusing during reconnection. Therefore, for events lasted more than one frames, only molecules localized within r$_{max}$ for at least two consecutive frames will be considered as bound molecules. Since we used relatively long acquisition time (500 ms) to blur the image of fast diffusing molecules, events that appeared in single frames were also taken into consideration as bound molecules to have a track length of 0.5 s. The duration of individual tracks (dwell time) was directly calculated based on the track length. We used 2 s as the time cutoff for mapping stable binding events.

MTT algorithm was used to track fast TF dynamics in the high excitation, fast acquisition (10 ms) condition. The resulting tracks were inspected manually by a homemade Matlab program. Tracks with incorrect linking events were discarded.

## 2D image registration, intensity map calculation, mask definition, and TF diffusion analysis

We took GFP-HP1 images before and after SMT experiment to make sure the cell nucleus and HC regions have not moved during the 5–6 min of single-molecule imaging. For experiment investigating co-localization of Pol II-enriched regions and Sox2 EnCs, image registration was performed by calculating and aligning nucleus outlines from both datasets. After background subtraction, the intensity map for heterochromatin regions in single cells was directly calculated by normalizing pixel intensity in the GFP-HP1 channel with the highest pixel intensity in the image. The intensity map for Dendra2 Pol II or stable Sox2 binding sites was calculated by 2D Gaussian kernel density function implemented by Matlab. Specifically, the density probability of X, Y localizations of stable binding events was evaluated in a 100 × 100 matrix with arbitrary units. The bandwidth for density estimation is 2 units. The resulting probability map was rescaled to the original image size. Composite images were constructed by superimposing the two intensity maps as two independent color channels. Binary mask for heterochromatin regions or enhancer clusters was calculated by applying a threshold cutoff of 0.2 to the intensity map. 2D single-molecule tracks were divided to track segments resided in the mask and outside of the mask. Track segments from each catalog were pooled. Diffusion coefficients were calculated from tracks with at least eight consecutive frames by the MSDanalyzer (*Tarantino et al., 2014*) with a minimal fitting $R^2$ of 0.8.

## 3D pair correlation function and calculation

According to *Peebles and Hauser (1974)*, we define the pair correlation function *g(r)* measures the probability *dP* of finding an enhancer site in a volume element *dV* at a separation *r* from another enhancer site.

$$\Delta P = n g(r) \Delta V, \tag{3}$$

where n is the mean number density of the enhancers in the nucleus.

In practice, the pair correlation function can be estimated from a sample of objects counting the pairs of objects with different separations r [Peebles & Hauser [4] estimator]:

$$g(r) = \frac{N_R}{N} \frac{DD(r)}{RR(r)}, \tag{4}$$

where *DD(r)* and *RR(r)* are counts of pairs of enhancers (in bins of separation) in the data catalog and in the random catalog, respectively.

The random catalog consists of uniformly distributed positions in the same volume defined by data catalog 3D convex hull. To reduce the noise, we computationally generate the random catalog that has a size 10 times greater than that of the data catalog. The normalizing coefficients containing the numbers of points in the initial (N) and random (Tarantino et al.) catalogs are included in the estimator.

Here, non-redundant pair wise Euclidean distance set within each catalog can be constructed by

$$d_{st}(i,j)(i \neq j) = \| \vec{r}_i - \vec{r}_j \|. \tag{5}$$

We define:

$$C(d_{st}(i,j),r) = \begin{cases} 0 \ (d_{st}(i,j) > r) \\ 1 \ (d_{st}(i,j) \leq r) \end{cases}. \tag{6}$$

The bin size of the g(r) distribution function is $\Delta r$.
Then,

$$g(r) = \frac{N_R}{N} \frac{DD(r)}{RR(r)} = \frac{N_R}{N} \frac{\sum_{i,j} C_{DD}\left(d_{st}^{DD}(i,j), r + \frac{\Delta r}{2}\right) - \sum_{i,j} C_{DD}\left(d_{st}^{DD}(i,j), r - \frac{\Delta r}{2}\right)}{\sum_{i,j} C_{RR}\left(d_{st}^{RR}(i,j), r + \frac{\Delta r}{2}\right) - \sum_{i,j} C_{RR}\left(d_{st}^{RR}(i,j), r - \frac{\Delta r}{2}\right)}. \tag{7}$$

DD(r) and RR(r) are calculated by pair wise distance function supplied in Matlab 2013a version with 50 nm as the histogram bin.

## 2D pair cross-correlation

For investigating the spatial cross-correlation between the localizations of two factors, we first converted the 2D super-resolution localization densities to image intensity maps via a 2D Gaussian kernel density function (see details in *Intensity Map Calculation, Mask Definition, and TF Diffusion Analysis*). Then, we implemented the Pair Cross-Correlation function using a well-established fast Fourier transform based method (*Veatch et al., 2012*). Specifically,

$$\text{Cross-Correlation Function, } c(\vec{r}) = Re\left\{\frac{FFT^{-1}\left(FFT(I_1) \times conj\left[FFT(I_2)\right]\right)}{\rho_1 \rho_2 N(\vec{r})}\right\}, \tag{8}$$

$$N(\vec{r}) = FFT^{-1}(|FFT(Mask)|^2). \tag{9}$$

The normalization fact $N(\vec{r})$ is the autocorrelation of a mask that has the value of 1 inside the nucleus region of the cell. The cell nucleus mask was obtained from the GFP-HP1 or Dendra2-Pol II wide-field image by intensity thresholding.

Here, *conj*[] indicates a complex conjugate. FFT and FFT$^{-1}$ were implemented by fft2() and ifft2() functions in Matlab. $\rho_1$ and $\rho_2$ are the average surface densities of images $I_1$ and $I_2$ respectively, and Re{} indicates the real part. Autocorrelation was calculated by using identical $I_1$ and $I_2$.

This computation method of tabulating pair cross-correlations is mathematically similar to brute force averaging methods. Correlation functions were angularly averaged using polar coordinates (Matlab command *cart2pol()*), and then binning by radius. Final values are obtained by averaging within the assigned bins in the radius. Because the intensity map pixel size is 160 nm after the 2D Gaussian kernel density estimation, we only calculated pair cross-correlation function at a range of diffraction limited radii (r > 160 nm). In this regime, over-counting has negligible effects on the final output of auto- or cross-correlation function.

Permutation was performed by randomizing pixels spatially within the nucleus mask for both images before calculating the cross-correlation.

## The fluctuation model of enhancer clustering

We extended previously published fluctuation model for measuring two dimensional heterogeneous distribution of membrane proteins to quantify 3D enhancer clustering (*Sengupta et al., 2011*).

Specifically,

$$G(r)_{observed} = G(r)_{stoch} + G(r)_{enhancer} \otimes G(r)_{PSF}, \tag{10}$$

$G(r)_{observed}$, the observed pair correlation function as calculated in the previous section. $G(r)_{stoch}$, the contribution of multiple appearances of the same molecule at a fixed site to the measured total correlation function. In our case, molecules are sparsely labeled. And, we track TF molecules through time/frames and, for each stable binding event, we only count once with the average localization over multiple frames. Thus, the contribution of $G(r)_{stoch}$ is negligible under this condition.

An exponential function can be used to approximate the correlation function of enhancers if they are present in randomly distributed clusters of no defined shape.

$$G(r)_{enhancer} = AExp\left(-\frac{r}{\varepsilon}\right) + 1, \tag{11}$$

A, the fluctuation amplitude which is in proportion to the ratio of the density of enhancers in clusters to the average density across the entire space. $\varepsilon$, the fluctuation range which is in proportion to the size of the clusters.

The correlation function of PSF of the imaging method is denoted as $G(r)_{PSF}$ and can be approximated by

$$G(r)_{PSF} = \frac{1}{8\pi^{\frac{3}{2}}\bar{\sigma}^3}Exp\left(-\frac{r^2}{4\bar{\sigma}^2}\right), \tag{12}$$

$\bar{\sigma}$, is calculated by $\bar{\sigma}^2 = \bar{s}^2 + \bar{a}^2\!\big/_{\!12}$, wherein $\bar{s}$ is the average s.d. of the PSF and $\bar{a}$ is the average voxel dimension.

Then, the final observed pair-correlation function can be fitted by the equation below:

$$G(r)_{observed} = \left(A Exp\left(-\frac{r}{\varepsilon}\right) + 1\right) \otimes \left(\frac{1}{8\pi^{\frac{3}{2}}\bar{\sigma}^3} Exp\left(-\frac{r^2}{4\bar{\sigma}^2}\right)\right),$$ (13)

$\otimes$, denotes convolution operator.

For the uniformly distributed, simulated sites, the data were fitted with $G(r)_{enhancer}$ directly.

Curve fitting is performed using the trust-region method implemented in the Curve Fitting Matlab toolbox.

## TF Brownian motion simulation

According to Einstein's theory, the mean square displacement of Brownian motion is described as

$$<r^2> = 2dDt,$$ (14)

$d$, dimensionality, in our case, $d = 3$. $D$, diffusion coefficient.

To computationally simulate Brownian motion in the Cartesian coordinate system, we uncoupled each jump to x, y, z one dimensional steps defined by the equation below.

$$\begin{pmatrix} x(t+\delta t) \\ y(t+\delta t) \\ z(t+\delta t) \end{pmatrix} = \begin{pmatrix} x(t) \\ y(t) \\ z(t) \end{pmatrix} + \sqrt{2D\delta t} \begin{pmatrix} N_1 \\ N_2 \\ N_3 \end{pmatrix},$$ (15)

where $N_i$ are independent random numbers obeying Gaussian distribution with a zero mean and a variance of 1 and $dt$ is the sampling interval.

## Monte Carlo simulation of target search in the nucleus

Simulation of target search was performed with MathWorks Matlab 2013a. The target search problem was reduced to random walk trapping problem with boundary and multiple traps. Specifically, we limited the 3D diffusion of the TF in a nucleus with a radius of 5 µm. Considering the average length of nucleosome depleted regions as 100–200 base pairs and the persistent length ($l_p$) of naked DNA as about 45 nm (135 bps) (**Williams and Maher, 2011**), target site radius was set as 40 nm. Overlaps between targets were not allowed in our simulation experiment. Specifically, the minimal distance (80 nm) allowed between the center of two targets is two times of the target radius (40 nm). In our simulation experiment, the TF binding probability to target is 1 when the TF reaches individual targets. The number of target sites was 7000 as estimated in our previous work. The mean diffusion coefficient (D) of the TF is 10 µm²s⁻¹; the sampling interval ($\delta t$) is 10 ms. Under this condition, the X, Y, Z step sizes are about 14 nm (when $N_i = 1$) much smaller than the target size, suggesting that the space is not under-sampled. We computationally manipulated the spatial distribution of target site and injection site position of the TF in the nucleus as indicated in the specific experiment and we recorded the first passage (3D) time and trajectory of each trial before the TF was reaching the first target according to the first-hitting-time model in the survival theory.

## Extra mathematic equations

Relative Fluorescence Intensity (RFI) for probing Sox2 levels in heterochromatin.

$$RFI = \frac{I_{Heterochromatin} - I_{Background}}{I_{Surrounding} - I_{Background}},$$ (16)

I, stands for mean gray intensity for the selected region.

Single-component exponential fitting of $\tau_{3D}$

$$Density(\tau) = e^{-\frac{\tau}{t}}, \tag{17}$$

t, the mean lifetime.

Two-component exponential fitting of $\tau_{3D}$

$$Density(\tau) = Fe^{-\frac{\tau}{t_1}} + (1-F)e^{-\frac{\tau}{t_2}}, \tag{18}$$

$t_1$, $t_2$ the mean lifetime for each component.

The relationship between enhancer concentrations and 3D time ($\tau_{3D}$).

According to the Smoluchowski Equation,

$$k_{on} = 4\pi RD, \tag{19}$$

R, capture radius. D, diffusion coefficient.

Observed on-rates for TFs are defined by the following equation,

$$k_{on}^* = k_{on}[DNA] = k_{on}\rho(r) = 4\pi RD\rho(r), \tag{20}$$

where enhancer concentrations ([DNA]) are a function ($\rho(r)$) of r relative to the center of the cluster.

This is the equation linking enhancer concentrations ($\rho(r)$) to $\tau_{3D}$.

$$\tau_{3D} = \frac{1}{k_{on}^*} = \frac{1}{4\pi RD\rho(r)} \tag{21}$$

## ChIP-exo library preparation

ES cells were treated with 50 nM TSA (Sigma-Aldrich, USA: T8552) for 8 hr. Then, cells were cross-linked by formaldehyde and harvested. Chromatin Immunoprecipitation (ChIP) was performed according to *Boyer et al. (2006)* with minor modifications. Briefly, cross-linked ESC chromatin was sheared using Covaris S2 system to a size range of 100 bp–400 bp. Immunoprecipitation was conducted with either specific antibody conjugated Protein A Sepharose beads (GE Healthcare). ChIP-exo library was prepared by following the published protocol with minor modifications (*Rhee and Pugh, 2011*). Specifically, we adapted the SoLid sequencer adaptors/primers to make the final library compatible with the illumina Tru-seq seq small-RNA system. Anti-Sox2 (R&D Systems, Minneapolis, MN Cat. # AF2018, Lot # KOY0112011) antibody was used for the ChIP experiment. The detailed primer information is in *Supplementary file 1*.

## ChIP-exo peak calling and bound-region definition

We sequenced exo libraries in 60 bp (Sox2 TSA) single-end format by using the illumina HiSeq platform. After removal of the 3' most 24 bp (Sox2 TSA) or 14 bp (Sox2 Wild type: 50 bp reads) which tend to have higher error rates, we mapped our sequencing data back to the mouse reference genome (mm10) by Bowtie 2 (*Langmead and Salzberg, 2012*). After mapping, we normalized the total mapped reads for each factor to 40 million. We further reduced the mapped read regions to single 5'-end point, which reflects the cross-linking point between protein and DNA. The resulting cross-linking point distribution was used to identify peaks on the forward (Left) and reverse (Right) strand separately using the peak calling algorithm in GeneTrack (*Albert et al., 2008*). For bound-region calculation, we first identified any pairs of left and right peaks that were located within 20 bps to each other. Then, we defined the window between the middle point of the left peak and that of the right peak as the bound-region. Peak-pairing and bound-region calculation were performed with Python programming (the script is available at https://github.com/Jameszheliu/PeakPairingProgram). Sox2 ChIP-exo sequencing data using wild type ES cells were obtained from GEO with the accession number of GSM1308179 (*Chen et al., 2014b*). Sox2 ChIP-exo data using TSA treated ES cells were deposited to NCBI GEO with the accession number of GSE62972.

## Acknowledgements

We thank Herve Rouault, Carl Wu, Xavier Darzacq, and Timothee Lionnet for proofreading the manuscript, Kai Wang for the optical schematics, C Morkunas and S Moorehead for general assistance.

# Additional information

## Competing interests

RT: President of the Howard Hughes Medical Institute (2009-present), one of the three founding funders of *eLife*, and a member of *eLife's* Board of Directors. The other authors declare that no competing interests exist.

## Funding

| Funder | Grant reference number | Author |
|---|---|---|
| Howard Hughes Medical Institute | Junior Fellow Program | Zhe Liu |
| Howard Hughes Medical Institute | Janelia Research Campus | Wesley R Legant, Bi-Chang Chen, Li Li, Jonathan B Grimm, Luke D Lavis, Eric Betzig, Robert Tjian |

The funders had no role in study design, data collection and interpretation, or the decision to submit the work for publication.

## Author contributions

ZL, Conception and design, Acquisition of data, Analysis and interpretation of data, Drafting or revising the article; WRL, B-CC, Acquisition of data, Analysis and interpretation of data; LL, Conception and design, Acquisition of data, Analysis and interpretation of data; JBG, LDL, Acquisition of data, Contributed unpublished essential data or reagents; EB, Conception and design, Acquisition of data; RT, Conception and design, Drafting or revising the article

# Additional files

## Supplementary file

• Supplementary file 1. The fluctuation model fitting results, localization parameters, and ChIP-Exo primers.

## Major datasets

The following dataset was generated:

| Author(s) | Year | Dataset title | Dataset ID and/or URL | Database, license, and accessibility information |
|---|---|---|---|---|
| Li L, Zhe L | 2014 | High-resolution Sox2 DNA-binding sites mapping by ChIP-exo in TSA-treated ES cells | http://www.ncbi.nlm.nih.gov/geo/query/acc.cgi?acc=GSE62972 | Publicly Available at NCBI Gene Expression Omnibus. |

The following previously published dataset was used:

| Author(s) | Year | Dataset title | Dataset ID and/or URL | Database, license, and accessibility information |
|---|---|---|---|---|
| Zhe L | 2014 | Sox2 | http://www.ncbi.nlm.nih.gov/geo/query/acc.cgi?acc=GSM1308179 | Publicly Available at NCBI Gene Expression Omnibus. |

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
