## [Decision Letter]

Thank you for sending your work entitled "3D Imaging of Sox2 Enhancer Clusters in
Embryonic Stem Cells" for consideration at *eLife*. Your article has been
favorably evaluated by Sean Morrison (Senior editor), a Reviewing editor, and 3
reviewers.

The Reviewing editor and the reviewers discussed their comments before we reached this
decision, and the Reviewing editor has assembled the following comments to help you
prepare a revised submission.

This is excellent work focusing on the 3D organization of enhancers into spatial
clusters and its consequences on Sox2 TF target search dynamics. Sox2 is one of the most
important pluripotent TFs that has been studied widely. However, due to its high copy
number and numerous targeting sites, it has not been possible to visualize its dynamic
organization, which is crucial for understanding how Sox2 differentiates its targeting
genes. The authors developed novel in-nucleus single molecule imaging approaches and
corresponding analyzing tool sets that are of general value for live cell single
molecule studies. In combination with numerical simulations and ChIP-exo mapping, the
authors came up with two important conclusions that are new for eukaryotic regulation of
gene expression.

The imaging is of high quality and the datasets are therefore robust. While the
clustering is obvious and its analysis relatively straightforward, the rational to
describe a switch in exploration modes within Sox2 clusters is questionable. The authors
show that binding is different within these two domains and it is not clear how this is
an indication of different diffusion modes. Different modes of exploration for
transcription factors have been recently described (24), and it would have been interesting to compare the Sox2
data to this study, especially explaining in more depth how the different modes of
exploration described here relate to each other and how the experimental methods leading
to these conclusions relate. So, while there is no doubt regarding the novelty and
general interest of this work, some of the conclusions are not well-supported by the
data presented, and the kinetics interpretation is not self-consistent. The four major
issues that need to be addressed are:

First, a major conclusion of this work is the clustering of Sox2 enhancers. However, the
authors provide little evidence to support the conclusion that the sites with >3
sec residence time are indeed enhancer elements. There are a large number of possible
explanations for this observation. For example, the long-lived sites could be regions
with a high density of nonspecific interaction partners. Alternatively, each site may be
an individual enhancer instead of a cluster of enhancers. In a related issue, the
authors state that "...enhancer clustering does not appear simply to result from
differential chromatin packaging..." and later they entertain two distinct models: a
polymer model and TF-directed chromatin looping. Given that Sox2 has thousands of
cognate sites, what fraction of Sox2 cognate sequences form clusters? If one can define
only two categories, clustered and scattered, what is the proportion of each type and
what are the functional implications for the sites that are scattered? To address these
inter-related issues, the author should determine whether Sox2 mutants that do not bind
to DNA specifically still form clusters. The authors should already have the mutant cell
lines as described in their Cell paper published this year. The identification of the in
vivo entities that they are imaging and tracking is key to their interpretations. So, a
mutant that can't bind specifically is an ideal control, and a reasonable control
to expect.

Second, there is no compelling evidence supporting the conclusion that "a 1D-search mode
predominates within individual clusters". The closest evidence for 1D-search comes from
a previously reported in vitro assay that is consistent with, but not directly
demonstrating, 1D diffusion (Chen et al 2014). The data presented in this work do not
address whether 1D-search occurs in vivo, let alone within individual clusters.
Furthermore, the search model and simulation presented in this work do not even include
1D diffusion. The simulation is thus inconsistent with the major claims. This
inconsistency may arise from the fact that the simulations that explain the trapping
behavior are not sufficiently described. The authors describe the binding sites as being
spheres with a 40 nm radius, and one could rationalize that in a clustering scenario
these spheres would create a continuous larger binding site. If this is the case, then
the effect observed can simply be explained using the Smoluchowski equation as stated by
authors. If the authors envision another effect involving a reduction in the search
dimensionality, then it is worth describing this effect and showing evidence for it.
Among questions important to discuss: why chose 40nm radius; once the binding sites are
found, is the binding probability 1 (if yes, why?)? The authors come back to the 3D
versus 1D scenarios and this comes in apparent contradiction to the previous
simulations, unless the 40 nm binding cross section already accounts sliding, in which
case the argument is circular. This part of the manuscript requires a much clearer
description of the model and the assumptions, as well as, a much better explanation of
the experimental evidence.

Third, the conclusion that Sox2 enhancers overlap with a subset of PolII enriched
regions is poorly justified. The strongest evidence is the small pairwise cross
correlation presented in Figure 4. However, as
both PolII and Sox2 are diminished in heterochromatins and nucleoli, the spatial
distribution is by default correlated, and so is the pixel-to-pixel correlation in the
supplement. The authors used permutated distribution to show that the correlation
between PolII and Sox2 is above background. But the permutation is only meaningful if it
also excludes heterochromatins and nucleoli. Clarification of this issue is also
needed.

Finally, fourth, what are the dynamic properties of EnCs? the authors state: "These
transiently formed EnC clusters ...", but there are no results that clearly describe the
dynamic behavior of EnCs in terms of their formation and disassembly, such as the
measurements of RPB1 dynamic clustering by [9]. Related to this question, the visualization of EnCs is achieved by
imaging enhancer-bound Sox2. The work beautifully measured the on-rate of Sox2 in the
clusters. How is the off-rate and how is it related to the cluster dynamics? In addition
to analyzing dynamics within and out of EnCs, it would be necessary to check if there is
anything special around the EnCs, as the chromatin structure may be different around the
EnCs compared with the bulk.

---

## [Author Response]

*First, a major conclusion of this work is the clustering of Sox2 enhancers.
However, the authors provide little evidence to support the conclusion that the sites
with >3 sec residence time are indeed enhancer elements. There are a large
number of possible explanations for this observation. For example, the long-lived
sites could be regions with a high density of nonspecific interaction partners.
Alternatively, each site may be an individual enhancer instead of a cluster of
enhancers. In a related issue, the authors state that "...enhancer clustering does
not appear simply to result from differential chromatin packaging..." and later they
entertain two distinct models: a polymer model and TF-directed chromatin looping.
Given that Sox2 has thousands of cognate sites, what fraction of Sox2 cognate
sequences form clusters? If one can define only two categories, clustered and
scattered, what is the proportion of each type and what are the functional
implications for the sites that are scattered? To address these inter-related issues,
the author should determine whether Sox2 mutants that do not bind to DNA specifically
still form clusters. The authors should already have the mutant cell lines as
described in their Cell paper published this year. The identification of the in vivo
entities that they are imaging and tracking is key to their interpretations. So, a
mutant that can't bind specifically is an ideal control, and a reasonable
control to expect*.

We thank the reviewers for this comment because there are several points that we should
clarify regarding this important issue:

We agree that it is presently not possible to prove that every Sox2 stable binding event
we detect occurs at a specific cognate binding site. However, the issue of specific
versus non-specific binding to chromatin/DNA was extensively addressed in our previous
Cell paper (7) and the relevant
results that provide strong evidence for the long-lived binding events to represent site
specific interactions are listed below:

1) We deleted the Sox2 DNA binding domain and tracked its movement. Removal of the DNA
binding domain resulted in the disappearance of the long-lived population from the dwell
time histograms of the truncated protein, suggesting that long-lived immobile particles
most likely correspond to Sox2 occupying specific cognate DNA target sequences.

2) Mutation of amino acids on the Sox2-DNA binding surface reduced the fraction and
lifetime of the long-lived population.

3) We reconstituted an in vitro purified TF system to study Sox2-DNA interaction
kinetics with surface-attached specific and non-specific DNA at single-molecule
resolution by TIRF. The analysis of the single-molecule data confirmed that the average
residence time of Sox2 on a DNA probe containing a canonical Sox2 binding site is 16.9 s
while the average residence time on a non-specific probe is 0.9 s. These numbers are
consistent with our “in vivo” residence time measurements - specific
(∼12s) and nonspecific (0.8s), thus providing an independent measure consistent
with the notion that our in vivo imaging analysis likely successfully resolved specific
from non-specific TF-DNA dissociation kinetics (residence time) in live cells.

4) We also tracked Sox2 binding behavior in 3T3 cells containing or lacking Oct4. With
Oct4 over-expression, we observed a significant increase in Sox2 specific residence
time. This result strongly suggested that we are indeed measuring the specific residence
times of Sox2 at cognate Sox2/Oct4 sites where Oct4 interacts with Sox2 and stabilizes
its binding to chromatin.

Taken together, these previous findings provide compelling support for the conclusion
that long-lived residence times we observe can be used to discriminate Sox2 specific
from non-specific TF binding at endogenous single copy genes. We believe that a more
rigorous test of site specific binding will be possible once we develop imaging methods
that can unambiguously detect, identify and track individual gene/promoter loci using
strategies such as CRISPR/Cas9 tagging of targeted genes/alleles in the genome in live
cells. Our groups at Berkeley and Janelia are actively pursuing this important goal but
technology for single gene tracking is proving to be challenging.

To further test whether the clustering behavior of Sox2 long-lived stable binding sites
might arise from non-specific Sox2 interactions in the cell, we specifically
investigated the clustering behavior of shorter-lived (< 3 s) Sox2 binding sites
that were originally filtered out in our stable binding site mapping experiments. If the
Sox2 stable binding sites only reflected random non-specific interactions, the
clustering behavior of short lived binding sites would be the same as that of the long
lived binding sites. However, in contrast to this prediction, shorter-lived Sox2 binding
sites showed dramatically decreased clustering as reflected in the greatly reduced
fluctuation amplitudes of the pair correlation function plots. Instead, we found the
shorter-lived Sox2 binding sites showed very little clustering and greatly reduced
fluctuation amplitudes of the pair correlation function curves (Figure 2—figure supplement 1 and Figure 7). We also note that in many cases, we
observed little or no clustering of short-lived Sox2 binding sites within the same
territories where longer-lived stable Sox2 binding site clusters can clearly be observed
(Videos 3 and 6). These results
suggest that the long-residence time filtering strategy that we deployed here likely
enriches for specific binding site signals above the background of non-specific
interactions consistent with what we observed previously (Chen et al., 2014).

There are 2 reasons that it is not possible for us to use Halo-Sox2-TAD protein to
perform the control experiments as the reviewers suggested: the majority of the Sox2-TAD
binding are very short-lived (less than 0.8 s) so that we cannot apply the 3 second
cutoff required for a reasonable comparison. In addition, the halo-Sox2-TAD protein is
highly unstable in ES cells and thus its expression levels are too low to be compatible
with the imaging strategy required to comprehensively map global binding sites. We have
now included these findings in the revised manuscript (Figure 2—figure supplement 1 and Video 6).

To determine whether the blinking of stably bound fluorescently tagged Sox2 molecules
significantly influences the observed “stable” binding of Sox2 in the
clusters, we plotted the number of detected spots as a function of frame number. These
plots show an initial decay that eventually reaches a plateau (Figure 2—figure supplement 2. Such a temporal decay
profile is more consistent with a bleaching dominant mechanism in which an equilibrium
state has been achieved between photo-bleaching and diffusive labeling. Moreover, Sox2
specific residence times (∼12s) are relative short compared to the total imaging
time (∼20 mins). Perhaps an even stronger argument that the Sox2 clustering
pattern we observe is not likely an artifact of the imaging modality can be derived from
the fact that imaging chromatin bound halo-H2B molecules failed to show such a prominent
clustering pattern. We have modified the revised MS accordingly to include this
important point.

In this paper, we implemented an imaging strategy by taking advantage of the previously
characterized dynamic in vivo Sox2-DNA binding properties (7) coupled with the recently developed lattice
light-sheet microscopy (6) and
improved fluorescent dyes (Grimm et al., Submitted) to systematically map globally where
the stable Sox2:DNA binding events occurs inside the 3D space of individual living ES
cells. The resulting 3D map is aimed at resolving where stable Sox2 bound enhancers are
located in the live cell nucleus rather than to precisely map all Sox2 cognate binding
sites and link these sites to specific sequences in the genome defined by ChIP-exo.
Clearly, that is a goal that we are working towards but have not yet achieved.
Currently, we can make a qualitative conclusion that a reduced Sox2 binding site
clustering in the linear genome after TSA treatment correlates with reduced Sox2 stable
binding site clustering in 3D. It would be premature for us to make quantitative
statements of exactly how many Sox2 binding sites are within each cluster and how many
binding sites are scattered outside of clusters. We expect that depending on different
cut-offs, analysis methods and sampling depths, the fraction of clustered sites versus
scattered sites could vary considerably. Much more theoretical work needs to be
conducted with the 3D stable binding site maps from different TFs to determine the
optimal cut-offs and methods that should allow us to better define
“clustered” versus “scattered” categories.

*Second, there is no compelling evidence supporting the conclusion that "a
1D-search mode predominates within individual clusters". The closest evidence for
1D-search comes from a previously reported in vitro assay that is consistent with,
but not directly demonstrating, 1D diffusion (Chen et al 2014). The data presented in
this work do not address whether 1D-search occurs in vivo, let alone within
individual clusters. Furthermore, the search model and simulation presented in this
work do not even include 1D diffusion. The simulation is thus inconsistent with the
major claims. This inconsistency may arise from the fact that the simulations that
explain the trapping behavior are not sufficiently described. The authors describe
the binding sites as being spheres with a 40 nm radius, and one could rationalize
that in a clustering scenario these spheres would create a continuous larger binding
site. If this is the case, then the effect observed can simply be explained using the
Smoluchowski equation as stated by authors. If the authors envision another effect
involving a reduction in the search dimensionality, then it is worth describing this
effect and showing evidence for it. Among questions important to discuss: why chose
40nm radius; once the binding sites are found, is the binding probability 1 (if yes,
why?)? The authors come back to the 3D versus 1D scenarios and this comes in apparent
contradiction to the previous simulations, unless the 40 nm binding cross section
already accounts sliding, in which case the argument is circular. This part of the
manuscript requires a much clearer description of the model and the assumptions, as
well as, a much better explanation of the experimental evidence*.

We agree with this assessment and have modified our discussion of this point
accordingly. To put this in context, we present some of our thoughts on this point.
Previous results from our in vitro single molecule experiments strongly suggest that
when colliding with DNA nonspecifically, Sox2 undergoes some degree of 1D sliding (7). However, it is important to
note that currently no direct assays are available to study in vivo TF-1D sliding on
DNA. Thus, we completely agree that we have not been able to prove 1D search in vivo
yet. To be conservative, in the revised manuscript, we have moved all the 1D searching
related speculation to the discussion and only suggest 1D searching as a possible
contributing mechanism.

Secondly, we need to clarify that in our numerical simulation experiments, we only
intend to test how target site clustering might affect the 3D diffusion first passage
times (τ_3D_). For this purpose, we define the “Fold of
Delay” as the average first passage time in the “clustered” case
divided by the averaged first passage time obtained in the “uniform” case.
Thus, because the binding probability (1 or not) when the TF reaches individual targets
is identical in both the uniform and the clustered cases, the trend that we observed for
“Fold of Delay” would be the same due to the normalization step. We have
emphasized this point in the revised manuscript and the reason for us to choose 40nm as
the target radius is based on DNA polymer model is now described in the text.

Finally, it is important to note that in our simulation experiments, one constraint for
generating targets is that there is no overlap between target sites. Specifically, the
minimal distance (80nm) allowed between the center of two targets is two times of the
target radius (40nm). However, we note that in this case two targets might still in
contact with each other on surface. To further eliminate the possibility that targets
might create a continuous larger binding site, we reduced to radius of targets to 30nm
while maintaining the minimal distance between the centers of two targets as 80nm. In
this case, there are no possible contacts between targets. Under this condition, we
essentially got very similar simulation results. Thus, it’s highly unlikely that
the effects of target site clustering on τ_3D_ that we observed was due
to target site fusion. The new simulation results have been included in Figure 6—figure supplement 2. As the
reviewer suggested, we have more explicitly described the parameters for simulation
experiments in the revised manuscript.

*Third, the conclusion that Sox2 enhancers overlap with a subset of PolII
enriched regions is poorly justified. The strongest evidence is the small pairwise
cross correlation presented in*
Figure 4*. However, as
both PolII and Sox2 are diminished in heterochromatins and nucleoli, the spatial
distribution is by default correlated, and so is the pixel-to-pixel correlation in
the supplement. The authors used permutated distribution to show that the correlation
between PolII and Sox2 is above background. But the permutation is only meaningful if
it also excludes heterochromatins and nucleoli. Clarification of this issue is also
needed*.

Thanks for this suggestion. We have conducted pixel-to-pixel correlation in masks
defined by Pol II high regions. Regions such as nucleoli and heterochromatin with low
Pol II levels were omitted in these masks. Significant positive correlations between
Sox2 EnC and Pol II persisted under this condition. The observed positive correlations
were diminished when we performed in-mask permutation experiments. These results are now
included in Figure 4—figure supplement 1.

*Finally, fourth, what are the dynamic properties of EnCs? the authors state:
"These transiently formed EnC clusters ...", but there are no results that clearly
describe the dynamic behavior of EnCs in terms of their formation and disassembly,
such as the measurements of RPB1 dynamic clustering by*
[9]*.
Related to this question, the visualization of EnCs is achieved by imaging
enhancer-bound Sox2. The work beautifully measured the on-rate of Sox2 in the
clusters. How is the off-rate and how is it related to the cluster dynamics? In
addition to analyzing dynamics within and out of EnCs, it would be necessary to check
if there is anything special around the EnCs, as the chromatin structure may be
different around the EnCs compared with the bulk*.

To study the dynamic properties of EnCs, we used the time-counting analysis method
developed by [9] to probe the
temporal profiles of arrival times of stable binding detections within individual
clusters. Interestingly, we didn’t observe significant bursting behaviors as
described for Pol II clusters. These results are consistent with a model that Sox2 EnCs
are relatively stable during the period (∼20mins) of the imaging acquisition.
Because Sox2 bound enhancers are chromatin based structures, it’s worth noting
that previous FRAP (Fluorescence recovery after photo-bleaching) experiments on core
histone components (27)
found that large-scale chromatin structures in live cells appeared stable with a
half-life of > 2∼4 hours which is much longer than the duration of our
imaging experiments. These findings together suggest that the enhancer clustering we
observed here likely accurately reflect the average 3D genome organization within
reasonably short spatiotemporal length scales. The new results have been included in
Figure 2—figure supplement 2.